# Biome changes in Asia since the mid-Holocene – an analysis of different transient Earth system model simulations

Anne Dallmeyer[1], Martin Claussen[1,2], Jian Ni[3,4,5], Xianyong Cao[4,6], Yongbo Wang[4,7], Nils Fischer[1], Madlene Pfeiffer[8], Liya Jin[9], Vyacheslav Khon[10,11], Sebastian Wagner[12], Kerstin Haberkorn[2], Ulrike Herzschuh[4,6]

[1]Max Planck Institute for Meteorology, Bundesstrasse 53, 20146 Hamburg, Germany
[2]Meteorological Institute, Centrum für Erdsystemforschung und Nachhaltigkeit (CEN), Universität Hamburg, Bundesstrasse 55, 20146 Hamburg, Germany
[3]Institute of Biochemistry and Biology, University of Potsdam, Maulbeerallee 3, 14469 Potsdam, Germany
[4]Alfred Wegener Institute Helmholtz Centre for Polar and Marine Research, Telegrafenberg A43, 14473 Potsdam, Germany
[5]State Key Laboratory of Environmental Geochemistry, Institute of Geochemistry, Chinese Academy of Sciences, Lincheng West Road 99, 550081 Guiyang, China
[6]Institute of Earth and Environmental Science, University of Potsdam, Karl-Liebknecht-Strasse 24-25, 14476 Potsdam, Germany
[7]College of Resource Environment and Tourism, Capital Normal University, Beijing 100048, China
[8]Alfred Wegener Institute Helmholtz Centre for Polar and Marine Research, 27568 Bremerhaven, Germany
[9]Key Laboratory of Western China's Environmental Systems, College of Earth and Environmental Sciences, Lanzhou University, Lanzhou 730000, China
[10]Institute of Geosciences, Christian-Albrechts Universität zu Kiel, Ludewig-Meyn-Str. 10-14, 24118 Kiel, Germany
[11]A. M. Obukhov Institute of Atmospheric Physics RAS, Pyzhevsky 3, 117019 Moscow, Russia
[12]Helmholtz Center Geesthacht, Institute for Coastal Research, 21502 Geesthacht, Germany

*Correspondence to*: Anne Dallmeyer (anne.dallmeyer@mpimet.mpg.de)

**Abstract.** The large variety of atmospheric circulation systems affecting the Eastern Asian climate is reflected by the complex  Asian vegetation distribution. Particularly in the transition zones of these circulation systems, vegetation is supposed to be very sensitive to climate change. Since proxy records are scarce, hitherto a mechanistic understanding of the past spatio-temporal climate-vegetation relationship is lacking. To assess the Holocene vegetation change and to obtain an ensemble of potential mid-Holocene biome distributions for Eastern Asia, we forced the diagnostic biome model BIOME4 with climate anomalies of different transient Holocene climate simulations performed in coupled atmosphere-ocean(-vegetation) models. The simulated biome changes are compared with pollen-based biome records for different key regions.

In all simulations, substantial biome shifts during the last 6000 years are confined to the high-northern latitudes and the Monsoon-Westerlies transition zone, but the temporal evolution and amplitude of change strongly depends on the climate forcing. Large parts of the southern tundra are replaced by taiga during the mid-Holocene due to a warmer growing season and the boreal treeline in northern Asia is shifted northward by approx. 4° in the ensemble mean, ranging from 1.5° to 6° in the individual simulations, respectively. This simulated treeline shift is in agreement with pollen-based reconstructions from northern Siberia. The desert fraction in the transition zone is reduced by 21% during the mid-Holocene compared to pre-industrial due to enhanced precipitation. The desert-steppe margin is shifted westward by 5° (1°-9° in the individual

simulations). The forest biomes are expanded north-westward by 2° ranging from 0°-4° in the single simulations. These results corroborate pollen-based reconstructions indicating an extended forest area in north Central China during the mid-Holocene. According to the model, the forest-to-non-forest and steppe-to-desert changes in the climate transition zones are spatially not uniform and not linear since the mid-Holocene.

## 1 Introduction

Eastern Asia ranges from the tropical to the polar climate zone and is affected by three major atmospheric circulation systems, i.e. the Indian monsoon, the East Asian monsoon system and the Westerly wind circulation (Fig.1). The different character and the interplay of these circulation systems as well as the connection to the tropical Pacific lead to strong climate variability and regionally very diverse climate conditions in Asia (e.g. Lau et al., 2000; Wang, 2006; Ding and Wang, 2008). This is furthermore intensified by the complex Asian orography including vast mountain ranges, high-elevated plateaus as well as deep depressions, basins and large plains (Broccoli and Manabe, 1992).

The regional climate peculiarities are reflected in the multifaceted vegetation distribution. The climate in the cold Arctic zones is too harsh for trees and this temperature limitation results in prominent transition of deciduous and evergreen taiga to tundra and even ice deserts in northern Asia (Larsen, 1980). In contrast, the warm tropical climate regions and large parts of East China are affected by the monsoon systems leading to high annual rainfall rates that support the growing of tropical and warm-temperate forests (Ramankutty and Foley, 1999). The monsoon-related diabatic heating and vertical uplift at the Himalayas induce strong subsidence of air north and west of the Tibetan Plateau leading to dry climate in Central and western Asia (e.g. Rodwell and Hoskins, 1996; Duan and Wu, 2005). Coincidently with this transition from monsoonal influenced climate to dry (Westerly wind dominated) climate, a transition zone of forest to steppe and steppe to desert forms in Central East Asia. Particularly in the two transition zones, i.e. the temperature-limited taiga-tundra margin and the moisture-limited forest-steppe-desert transition area, vegetation is supposed to be very sensitive to climate change and strong feedbacks are expected in case of climate and vegetation shifts due to large environmental and biophysical changes (e.g. Feng et al., 2006 and references therein).

During the mid-Holocene, cyclic variations in the Earth's orbit around the sun led to substantial changes in climate (Wanner et al, 2008). The monsoon circulations were enhanced (Kutzbach, 1981; Shi et al., 1993; Fleitmann et al., 2003; Wang et al., 2005) and the high northern latitudes experienced a much warmer summer climate (Sundqvist et al., 2010; Zhang et al., 2010). Pollen-based reconstructions for the mid-Holocene time-slice show an influence of these climate changes on the vegetation distribution. The northern treeline was shifted poleward accompanied by an expansion of the boreal forest

(MacDonald et al., 2000; Bigelow et al., 2003). The forests in East Asia were extended, the steppe-forest margin moved by about 500 km to the north-west (Yu et.al., 2000; Ren and Beug, 2002; Ren, 2007; Zhao et al.,2009) and the desert area in China and Inner Mongolia was substantially reduced compared to today (Feng et al., 2006; Zhao et al., 2009). However, in the complex environment of Asia, the locally limited reconstructions may not portray the general vegetation change. Particularly in the monsoon margin region, the major patterns of Holocene vegetation trends are poorly understood and documented (Zhao and Yu, 2012). Maps of reconstructed vegetation based on (few) individual site records have only been prepared for single key-time-slices (e.g. Yu et al., 2000). Vegetation trends - so far - have mostly been presented by trends of different taxa for single sites not depicting the regional vegetation trend appropriately.

Large climate model intercomparison projects have been established to assess the mid-Holocene to pre-industrial climate change and to validate the state-of-the-art climate models against the mid-Holocene climate (e.g. Joussaume et al., 1999; Braconnot et al., 2007a, b). Some of these models only poorly reflect observed Asian climate (Zhou et al., 2009 and references therein) and, accordingly, the simulated Holocene climate strongly deviates among the models and from reconstructions. Current Earth System models generally include interactively coupled dynamic vegetation models that are based upon comparable routines but differ in their individual parametrisations. In these routines, vegetation is commonly described by few plant functional types that may not be able to represent the diverse taxa found in Asia. Hence, it is impossible to partition the differences between modelled vegetation into those originating from different climates and those being related to the specific vegetation module configurations.

In this study, we therefore go one step back and re-analyse the Holocene vegetation change in Asia by using a diagnostic vegetation model, i.e. the biome model BIOME4 (Kaplan, 2001; Kaplan et al., 2003; Tang et al., 2009). This model has been used before to assess the Holocene biome shifts in the Arctic (Kaplan et al., 2003), in the northern hemisphere extra-tropics (Wohlfahrt et al., 2008), in the southern tropical Africa (Khon et al., 2014) and on the Tibetan Plateau (Song et al., 2005; Ni and Herzschuh, 2011; Herzschuh et al., 2011) by comparing time-slice simulations for the mid-Holocene and pre-industrial climate. Additionally, the sensitivity of vegetation distribution simulated by BIOME4 to different present day input climatologies has been tested for Eastern Asia (Tang et al. 2009). But, to date BIOME4 has not been applied for the entire Asian region.

To get a systematic overview on the spatial pattern of Holocene vegetation change in Asia, we force BIOME4 with climate anomalies from different transient Holocene climate simulations performed with coupled atmosphere-ocean models that partly also include dynamic vegetation models. With this setup, we allow not only for the comparison between the mid-Holocene and the pre-industrial biome distribution, but also for the analysis of Holocene vegetation changes.

95 The main aims of this study are: a) to get a consistent ensemble of possible changes in biome distribution since the mid-Holocene, b) to test the robustness of the simulated vegetation changes and quantify the differences among the models, i.e. to assess how large the vegetation variability is that results from different climate forcings, and c) to compare simulated vegetation changes in selected key regions with pollen-based reconstructions.

## 2 Methods

### 2.1 Vegetation model: BIOME4

BIOME4 (Kaplan, 2001; Kaplan et al., 2003) is a terrestrial biosphere model that calculates the global equilibrium biome distribution based on a prescribed climate, taking biogeographical and biogeochemical processes into account.

Basic input variables are monthly mean climatologies of temperature, cloud cover (sunshine) and precipitation as well as the absolute minimum temperature, the atmospheric $CO_2$-concentration and soil physical properties such as the water holding capacity and percolation rates. These soil properties are based on global soil maps provided by the Food and Agriculture Organisation (FAO, 1995). The model distinguishes 13 different plant functional types (PFTs) that are defined by physiological attributes and bioclimatic tolerance limits such as heat, moisture and chilling requirements or the cold resistance of the plants (Tab. 1). These limits determine the area where the PFTs could exist in a given climate. Competition between the calculated PFTs is incorporated by ranking the PFTs according to their simulated relative net primary productivity (NPP), leaf area index (LAI) and the mean soil moisture (Kaplan et al., 2003). The biomes are then identified based on the dominant and second dominant PFTs. The original BIOME4 version includes 28 different biomes. To better track main biome shifts throughout the Holocene, we further grouped them into 12 mega-biomes (Tab. 2).

BIOME4 has originally been designed and calibrated to resolve global vegetation distributions under modern atmospheric $CO_2$-concentrations (325 ppm). To give consideration to the lower atmospheric $CO_2$-level during the Holocene as well as to the biome diversity and unique environment formed in the complex topography of Asia, we slightly modified and recalibrated the model. In detail, the following changes have been undertaken:

1. We implemented dependence of bioclimatic limits on the orography to better represent the vegetation in high-elevated areas such as the flanks of the Tibetan Plateau. In general, the bioclimatic limits given in the global BIOME4 model represent the horizontal biome differentiation (e.g. taiga and tundra, i.e. the northern treeline), but the limits in the vertical (e.g. the upper treeline) differ. Limits were taken from BIOME4-TIBET (Ni and Herzschuh, 2011). We choose the ETOPO5

dataset (Data Announcement 88-MGG-02: Digital relief of the Surface of the Earth. NOAA, National Geophysical Data Center, Boulder, Colorado, 1988) as orography.

2. BIOME4 performed poorly when simulating the Asian steppe distribution (cf. Tang et al, 2009). To address this limitation, we added a rain limit for vegetation that is conform with observations and operated well in test runs. In warm regions (growing degree days (GDD5) exceed 1200 °C) with less than 400 mm of rain per year, steppes are preferred. In warm regions with less than 200 mm/year, deserts prevail (Pfadenhauer and Klötzli, 2014).

3. We re-calibrate the model for pre-industrial $CO_2$-concentration (280 ppm) and the new reference climate data, the University of East Anglia Climatic Research Unit Time Series 3.10 (CRU TS3.10, University of East Anglia, 2008, Harris et al., 2012), i.e. we slightly modified the LAI and NPP constraints in the PFT and biome assignment to better match the observed vegetation distribution (see Fig. A1 in the Appendix). The new LAI and NPP limits are within the range of observations, but LAI and NPP measurements strongly vary and profound and comprehensive dataset do not exist for the Asian region.

The difference between the modified and original BIOME4 model can be seen in the Appendix (Fig. A2) based on the pre-industrial reference simulation and the ensemble mean simulation for the mid-Holocene (including comparison with reconstructions).

## 2.2 Reference Simulation 0k

As reference simulation for the modern biome distribution (named pre-industrial or 0k in the following), we forced BIOME4 with the modern monthly mean climatology (1960-2000) taken from the University of East Anglia Climatic Research Unit Time Series 3.10 (CRU TS3.10, University of East Anglia, 2008, Harris et al., 2012) providing a more reliable climate background than pre-industrial climate reconstructions or simulations. Though, we prescribed pre-industrial atmospheric $CO_2$-concentration (280 ppm) to be consistent to the transient mid-Holocene to pre-industrial climate simulations and to partly come up with the fact, that modern vegetation is supposed to be not in equilibrium with the fast changing atmospheric $CO_2$-level. The differences between the reference simulation using 280 ppm and a simulation prescribing 360 ppm can be seen in the Appendix (Fig. A3).

Furthermore, the CRU TS3.10 data is provided in relatively high spatial resolution of 0.5°x0.5°, resolving the climate gradients along the complex Asian orography better than the Global climate model simulations.

The simulated modern biome distribution in Asia is shown in Fig. 2. For comparison, the modern potential vegetation distribution described in the Vegetation Atlas of China (Hou, 2001), the Digital Atlas of Mongolian Natural Environments (Saandar and Sugita, 2004) and the RLC Vegetative Cover of the Former Soviet Union data-set (Stone and Schlesinger, 2003) was translated into biomes (cf. Ni et al., 2000; Ni and Herzschuh, 2011). The model is able to represent the large-scale biome distribution in Asia and most biome transitions agree well with the reference. The main discrepancies of the simulated distribution and the reference occur in East Siberia, where the model predominantly suggests tundra as main biome instead of deciduous taiga. The Central Asian steppe is less extended and the cool/cold forest and evergreen taiga range too far northward in the model compared to the reference. Furthermore, the biome distribution on and around the Altai mountains are only partly resolved in the model. However, as the diverse taxa in Asia cannot always be assigned clearly to the limited number of biomes, the transition of similar biomes such as taiga and cool/cold forests may not be precisely definable and may therefore also not be represented well in the biome reference. Particularly the non-forest biomes, e.g. different herbs, cannot properly be represented by the limited number of PFTs used in BIOME4. The model cannot distinguish biomes sharing similar PFTs under similar bioclimatic conditions, which is particularly problematic for describing different types of tundra and mountainous vegetation (Ni and Herzschuh, 2011). In addition, bioclimatic constraints are defined to characterise global vegetation and are therefore too broad to represent all regional vegetation types appropriately, even though we adapted the climate limits and biome assignment rules. Furthermore, better resolved soil properties datasets are needed to improve the vegetation simulation.

One additional factor complicating a proper evaluation of the BIOME4 model under present day climate conditions relates to the scarcity of observed and quality controlled meteorological observations. Even the CRU data set provides a good estimate over data-rich areas the underlying algorithm used for producing the gridded data set cannot compensate for regions with little or no observational data entering the data set.

## 2.3 Transient climate forcing data for the Holocene

To assess the Holocene vegetation changes in Asia, we drive BIOME4 with output of five different transient climate model simulations performed in a wide spectrum of fully-coupled atmosphere-ocean-vegetation models. Although, most of the models integrate versions of the ECHAM atmosphere model, the oceanic models are different leading to different climates in the here used simulations.

All climate models have been run into quasi-equilibrium under early- or mid-Holocene orbital conditions. Afterwards, the orbital parameters (Berger, 1978) were continuously being changed until pre-industrial (0k) conditions were reached. We started our analysis with the mid-Holocene time-slice, i.e. 6000 years before present (henceforth referred to as 6k). Atmospheric composition has been kept constant at pre-industrial values with $CO_2$-concentration set to 280 ppm in most models. Since the absolute minimum temperature (Tmin) was not provided by all models, we consistently calculated Tmin from the mean temperature of the coldest month as described in Prentice et al. (1992). The models have been tested to capture the present day Asian climate, in particular the Asian monsoon climate (cf. Dallmeyer et. al., 2015). A short overview of the different climate simulations is given below and summarised in Table 3. For details, we refer to the given references.

### 2.3.1 COSMOS and COSMOSacc

COSMOS is the model of the Community Earth System models network, initiated by the Max Planck Institute for Meteorology. The model consists of the atmospheric general circulation model ECHAM5 (Roeckner et al., 2003) coupled to the land-surface and vegetation model JSBACH, (Raddatz et al., 2007) and the Ocean model MPIOM (Marsland et al., 2003), and the Ocean-Atmosphere-Sea Ice-Soil coupler (OASIS3, Valcke et al., 2003; Valcke, 2013), which enables the atmosphere and ocean to interact with each other. JSBACH includes the dynamic vegetation module of Brovkin et al. (2009). ECHAM5 ran with a spectral resolution of T31L19, which corresponds to a longitudinal distance of approx. 3.75° and 19 levels in the vertical. The MPIOM was employed at a horizontal resolution of GR30 (formal resolution of ~3° x 3°) with 40 vertical levels. In this setup, two simulations have been performed, one with an accelerated orbit and one without acceleration. In the accelerated simulation (COSMOSacc, Varma et al., 2012; Pfeiffer and Lohmann, 2013), the orbital forcing was accelerated by a factor of 10, so that the Holocene is represented by only 600 instead of 6000 model years. In the non-accelerated simulation (COSMOS, Fischer and Jungclaus, 2011), orbital forcing was applied on a yearly basis. Atmospheric composition was fixed to pre-industrial values in both simulations.

### 2.3.2 ECHO-G

ECHO-G (Legutke and Voss, 1999) is an atmosphere-ocean general circulation model. The atmosphere is represented by ECHAM4 (Roeckner et al., 1996), which ran in the numerical resolution T31L19. The ocean is represented by HOPE-G (Wolff et al., 1997), that has an effective horizontal resolution of 2.8° x 2.8° using grid refinement in the tropical regions. In the transient simulation analysed in this study, non-accelerated orbital forcing was applied (Wagner et al., 2007). The

atmospheric composition was fixed to pre-industrial level. No interactive vegetation changes were used in this model simulation, i.e. the model is run with constant pre-industrial vegetation cover.

### 2.3.3 KCM

The Kiel Climate Model (KCM, Park et al., 2009) consists of the atmospheric general circulation model ECHAM5 (Roeckner et al., 2003) coupled to the ocean-sea ice general circulation model NEMO (Madec, 2006). ECHAM5 ran in the numerical resolution T31L19 corresponding to 3.75° on a great circle. NEMO ran with a horizontal resolution of approx. 2°x2° with increased meridional resolution of 0.5° close to the equator. The transient simulation analysed in this study covers the last 9500 years prescribing orbital forcing only (Jin et al., 2014). The change in orbital parameter was accelerated by a

factor of 10. Greenhouse gas concentration was kept constant on pre-industrial values.

### 2.3.4 PLASIM

PLASIM is an atmosphere model of medium complexity (Fraedrich, 2005) and ran with a horizontal resolution of T21 (approx. 5.6°x5.6° on a Gaussian grid) and ten levels in the vertical. In the simulation used for this study, PLASIM was coupled to the ocean model LSG (Maier-Reimer et al., 1993) and the vegetation module SimBA (Kleidon, 2006). The

transient simulation was not accelerated (Haberkorn, 2013), i.e. orbital forcing was applied on a yearly basis. In contrast to most other simulations, Holocene variations in atmospheric $CO_2$-concentration were additionally prescribed (from Taylor Dome, Indermuhle et al., 1999). Atmospheric $CO_2$-content was changed from 265-269 ppm during mid-Holocene to pre-industrial level (280 ppm). As the forcing related to the increase in $CO_2$ is relatively small compared to the orbital forcing, we do not expect any substantial impact of the non-constant $CO_2$ on the results drawn in this study.


### 2.4 Calculation of Biome distributions for the Holocene

The biome distributions are calculated for time-slices throughout the last 6000 years with an interval of 500 years. We used an anomaly approach to minimise the influence of systematic biases in the climate simulations on the vegetation distributions (cf. Wohlfahrt et al., 2008). For this purpose, the climate model outputs have been interpolated linearly to the

0.5° grid used in the CRU TS3.10 climate reference data. Then, the absolute differences between the monthly mean climatologies (long-term averages of 120 years, e.g. year 1-120 (1-12) of each model simulation for 6k, year 501-621 (51-62) for 5.5k, etc., the last 120 (12) years for 0k) simulated for each time-slice and the simulated pre-industrial climate have been added to the reference dataset. Negative values in precipitation or sunshine resulting from a too large negative

15                                                                                                 8

difference of the variables between 6k and 0k, compensating the present day values, have been set to zero. This anomaly

approach has the advantage of preserving regional climate pattern although the complex Asian orography is not resolved in the coarse spatial resolution used in the climate simulations (Harrison et al., 1998). We are however aware of the simplifications inherent to this approach in interpolating coarsely resolved GCM output (resolution approx. 3.75° and coarser) onto higher target grids (here 0.5°) without taking into account potentially important factors that lead to local variations in climate, such as changes in variability and feedbacks from the local to the meso- and large scale. Considering

the general character of our study this approach should however compensate for some of the GCM-related shortcomings related to orography and according processes. Atmospheric $CO_2$-concentration has been fixed to pre-industrial value (280 ppm) in all biome simulations.

To assess the climate variables being responsible for the biome shifts in the model since the mid-Holocene, we furthermore conduct a sensitivity study. We repeat the biome simulations for pre-industrial, but successively replace the climate variables

in the CRU TS3.10 input dataset with the respective variable of the simulated mid-Holocene climate. For instance, to test the sensitivity of the Eastern Asian biome distribution to the mid-Holocene precipitation change, we create an input dataset containing the pre-industrial 2m-temperature, cloud cover and absolute minimum temperature of the CRU TS3.10 dataset, as well as the simulated mid-Holocene precipitation.

To facilitate the discussion of the results, the biome simulations are named after the climate model simulation serving as

input for the BIOME4 model, e.g. the biome simulation forced with the climate calculated in COSMOS is referred to COSMOS in the following.

### 2.5  Pollen-based biome reconstruction for key areas of vegetation change

The simulated biomes were evaluated for key biome transition areas showing the strongest biome change in the model ensemble, i.e. the taiga-tundra transition zone in the high northern latitudes and the forest-steppe-desert-transition zone in

north-central China. Since the model fails to appropriately represent modern biome distribution in eastern Siberia (cf. Fig.2), the taiga-tundra key transition zone was confined to the north-central Siberian region.  For comparison, representative, high quality (with respect to dating and the data) pollen records covering the last 6000 years have been chosen. For the taiga-tundra-transition zone, a record from a small lake located on the southern Taymyr Peninsula (technical name: 13-CH-12; Klemm et al., 2016) is used, that is in line with the vegetation trend seen at other records located at the Siberian treeline

(Pisaric et al., 2000, McDonald et al., 2000, Bigelow et al., 2003). The biome change in the forest-steppe-transition zone is reflected by the record from Daihai Lake in Inner Mongolia (China, 40.5°N; 112.5°E, 1225 m a.s.l.; Xu et al., 2010) that is in

line with other records in north central China (Zhao et al., 2009 and references therein). To better compare the simulated biome distribution, these records were biomised in accordance with the BIOME4 biome classification. We applied the standard biomisation procedure to assign pollen taxa first to plant functional types (PFTs) and in a second step PFTs to biomes (Prentice et al, 1996) through a global classification system that was specified for Eastern Asia in Ni et al. (2014), but excluding anthropogenic PFTs. Finally, for each pollen spectrum in a record an affinity score for each biome is calculated. It is assumed that the biome with the highest score dominates in the pollen-source area of the lake, while a relatively lower score indicates less occurrence of a biome in the area. Accordingly, only the dominant biome types and the trends of different biomes but not their absolute coverage can be compared to model-based results. Please notice, that the pollen reconstructions are dated in calibrated years before present, i.e. before the year 1950 AD (cal. ka BP), thus the time-step 0 cal. ka BP is not identical with the time-slice '0k' used in the modelling result (i.e. a mean of 120 years).

## 3 Results

### 3.1 Simulated mid-Holocene biome distributions

Figure 3 shows the mid-Holocene biome distribution corresponding to the different climate simulations. Overall, the vegetation change is small and similar for all models. The main biome shifts occur in the high northern latitudes and in the transition zone of desert, steppe and forests in East Asia (95-125°E, 32-52°N).

Compared with the pre-industrial situation, deciduous taiga and boreal woodland further penetrate northward during mid-Holocene and the boreal tree line is shifted to the north by about 3-5° in the ensemble mean (for simplification, only ° is used for geographical distances given in degrees of latitude or longitude in the following). This shift ranges from approximately 1-2° in PLASIM to 4-6° in COSMOSacc.

The desert-steppe margin in East Asia is located further west by approx. 4° in the ensemble mean, substantially reducing the desert area in East Asia at mid-Holocene. The maximum response is found in COSMOSacc (approx. 9°), the weakest signal is shown by PLASIM (approx. 1°). The cool/cold forest and taiga biomes extend further north-westward into the pre-industrial East Asian steppe at 6k (approx. 2° in ensemble mean). The largest change occurs in COSMOS (3-4°). The climate change in PLASIM is too weak to induce a shift of the forest-steppe border in the southern part of the transition zone, north of 41° the steppe expands to the east.

Figure 4 displays the total change in area covered by the different biomes, averaged over all simulations and the entire Asian region (60-180°E, 15-80°N). In the ensemble mean, BIOME4 suggest a general increase in tropical and temperate forests by

about 11% and 38%, respectively, of the modern percentage of area during mid-Holocene. In contrast, warm and cool/cold forests are reduced by 14% and 20%, respectively, but the standard-deviation of the ensemble members is as large as the mean change. Evergreen (6%) and deciduous (27%) taiga strongly expand at the expense of tundra that experienced a reduction in area by 35% during mid-Holocene. The total change of shrubland and savannah/woodland is small. The area of grassland is increased by approximately 28% at mid-Holocene compared to the modern distribution. The desert fraction is reduced by 14%.

## 3.2 Climate factors determining the Eastern Asian vegetation change

The Holocene changes in bioclimate are discussed in the Appendix (Fig. B1). To assess the climate variables being responsible for the biome shifts in the model since the mid-Holocene, climate variables of the pre-industrial input dataset are gradually replaced by the respective variable of the simulated mid-Holocene climate. The results of this sensitivity study are shown in Fig. 5. As expected, most biome shifts between mid-Holocene and pre-industrial can be related to changes in temperature or precipitation. The prolongation of the warm season, associated with the general warmer summer climate in the northern latitudes (cf. Fig. B1) leads to the northward expansion of boreal forest. The biome shifts at the forest-steppe-desert transition in East Asia can already be induced by the mid-Holocene to pre-industrial precipitation change. The cloud cover (sunshine) plays a minor role and is mainly relevant in the differentiation of cold/cool forest and taiga (e.g. in COSMOSacc). The absolute minimum temperature has mainly an impact in the tropical region, where the decreased cold season temperature during the mid-Holocene leads e.g. to a slight southward shift of the tropical forest belt.

## 3.3 Transient biome shifts at the forest-steppe-desert transition zone in north-central China

According to the pre-industrial biome simulation, the vegetation in the forest-steppe-desert transition zone (95-125°E, 32-52°N) is dominated by forest biomes (45%) and to a lesser extend by grassland and desert biomes (18% and 26%, respectively). Tundra biomes cover only 10% of the region (Tab.4). Averaged over the area, the pre-industrial desert-steppe margin is located at 113°E and the forest-steppe margin at 120°E. This distribution has changed in the last 6000 years, probably due to differences in precipitation (cf. Fig 5). Figure 6 shows the simulated area mean and the meridional mean change in annual precipitation since the mid-Holocene relative to the modern precipitation (ensemble mean). Precipitation is increased by approx. 8%  on average and even by more than 15% in a broad region directly east of the pre-industrial desert-

steppe margin (approx. 113°E in the mean). In this region, rainfall stays on a relatively high level until 4.2k and eventually starts to decrease gradually, reaching modern climate conditions around 1.5k. This induces a reduction of the desert and a shift in the mean desert-steppe border. During the mid-Holocene, the area in the transition zone covered by desert biomes is declined by approx. 21% in the ensemble mean, ranging from a decreased fraction by 42% at mid-Holocene in one

simulation and a similar desert fraction as in modern climate in another simulation (Fig.7).

Averaged over all simulations, the mean desert border is located at 108°E during the mid-Holocene staying relatively constant until 4.5k. Afterwards, a shift of about 100 km to the east within 500 years occurs. The desert border stays relatively constant again until 2k and then moves gradually to its modern position. This gradual decline is interrupted by a period with strongly increased desert fraction compared to the millennia before at 1.5k.

On regional scale, most simulations reveal a substantially reduced desert fraction in large parts of the transition zone. During the entire period, the desert fraction is substantially reduced in a broad band between 102°E-115°E. Directly at the modern desert border, the vegetation changes at most with a decreased desert fraction of more than 20% during mid-Holocene (absolute values). But, the amplitude of the signal and the temporal change varies among the different models also on regional scale (cf. Fig. B in the Appendix). For instance, within Plasim, the steppe extent is relatively constant. Whereas the

simulation forced by COSMOSacc climate reveals the strongest response. The desert fraction is increased by more than 20% in large parts of the region from 6k to 2k within this simulation and the mean desert border is located at 104°E at 6k. Using the accelerated climate simulations as forcing, the desert distribution shows a strong variability. The generally increased precipitation during mid-Holocene favours the growing of forest in the transition area. Similar as for the desert, the different simulations reveal strong discrepancies among each other regarding the strength and pattern of forest cover change during

the last 6000 years (Fig. 8 and Fig. C2 in the Appendix). In most simulations, the forest fraction is increased in large parts of the region during the mid-Holocene, but they also suggest several periods with less forest than in other periods, at least on a local scale, e.g. between 3k and 3.5k and around 1.5k in the ensemble mean. The simulations based on the accelerated climate models reveal a much stronger response to the climate forcing than the other simulations.

Averaged over all simulations, the area in the transition zone covered by forest biomes is increased by approx. 8% during the

mid-Holocene compared to the modern fraction (Fig. 8). The maximum forest extent during the last 6000 years is reached at 5k and 2.5k with an increase in forest fraction by 12% compared to the modern distribution. However, the relatively small expansion of forest during the mid-Holocene is deceptive, because BIOME4 tends to replace taiga and cold forest with grass at the northern part of the transition zone and east of the Tibetan Plateau (100-110°E) due to higher temperatures at 6k. This tendency becomes particularly clear when looking at the simulated meridional mean change in forest cover since the mid-

Holocene (Fig. 8). In all simulations, forest cover is suggested to be decreased in the region between 103-110°E at 6k. Averaged over all simulations, this period of reduced forest areas lasts until 4k, ranging from 5k in COSMOSacc to 0.5k in ECHO-G.

In the region around 100°E, all simulations show an increase of the forest fraction (10% in the ensemble mean), which can mainly be attributed to an expansion of taiga on the Altai mountains and on the Tibetan Plateau. The forest cover gradually

decreases and reaches modern conditions at 2k in most simulations.

Directly west of the modern mean forest border (120°E), the strength of the signal strongly varies among the models. In the ensemble mean, large parts of the area experience a rise in the forest fraction by more than 10% during the mid-Holocene leading to a westward shift of the mean forest-steppe border by 2°. In this region, COSMOS and KCM reveal the strongest change with an increased forest fraction by more than 20% and a westward shifted forest-steppe-border by approx. 3° at 6k.

However, PLASIM suggests a relatively weak forest response to the Holocene climate forcing. The forest fraction is increased between 113°E and 118°E from 6k to 1k, but decreased eastward of 118°E by more than 10% during the entire period, actually resulting in an eastward shifted mean forest-steppe-border.

### 3.4  Transient biome shift in the north-central Siberian taiga-tundra transition zone

According to the pre-industrial biome simulation, the north-central Siberian taiga-tundra transition zone (66-120°E, 60-80°N) consists to 27% of tundra and to 73% of taiga (Tab. 6). The mean taiga-tundra border is located at 67.5°N. The vegetation in the high northern latitudes is particularly sensitive to changes in temperature (cf. Fig. 5). In the area mean and averaged over all simulations, the temperature is increased by approx. 1 K during the mid-Holocene, ranging from 1.9 K in ECHO-G to 0 K in PLASIM (Fig. 9). The climate in the northern parts of the region, i.e. north of ca. 71°N, stays relatively

warm until 4.5k showing   increased temperatures by more than 0.75 K in the ensemble mean. This warm period is interrupted several times by slightly colder periods. The southern part stays warm (>0.5 K) until 2.5k. Afterwards, the climate in the north-central taiga-tundra margin is similar to pre-industrial. These changes in temperature lead to an increase in forest cover and  a northward shift of the taiga belt.

All simulations show a substantially increased extent of taiga forest during the mid-Holocene and in the following millennia

(Fig.10 and Fig. C3 in the Appendix).  In the ensemble mean, the mean taiga border is located at 71.5°N at 6k with some extension to 73°N. In large parts of the area, the tundra fraction is reduced by more than 25% at mid-Holocene, directly at

the taiga border by even more than 50% (absolute values). The areal mean tundra fraction is decreased by 64% at 6k relative to the 0k extent (Fig. 10).

Averaged over all simulation, the taiga retreats relatively fast southward until 4k (back to 68.5°N) and then moves gradually and slowly back to the modern position. Nevertheless, the north-central Siberian taiga-tundra transition zone is characterised by a substantially increased taiga fraction during most of the period, staying on a high level until 1.5k.

The strongest signal is simulated in COSMOSacc with a reduced tundra area by 75% at 6k relative to the 0k coverage and a northward shifted taiga-tundra margin by 6°. In this simulation, The variability in tundra fraction is high, revealing periods with substantially decreased tundra area compared to other time-slices at 4k and 2.5k and a strongly increased tundra fraction north of 70°N at 1.5k. PLASIM reveals the weakest signal with a reduction by 48% and a shift of the taiga-tundra margin by only 1.5° during the Holocene.  Interestingly, COSMOS reveals a period with substantially reduced taiga fraction around 4k (up to 25%), resulting in a southward retreat of the taiga border to 68°N.

### 3.5  Pollen-based biome reconstruction for the forest-steppe transition and the taiga-tundra zones

Fig. 11 shows the reconstructed biome change based on two pollen records representing the vegetation change in the key-regions experiencing most pronounced changes in the biome simulations, i.e. the forest-steppe-desert transition zone in north-central China (represented by Lake Daihai record) and the taiga-tundra transition zone in north-central Siberia (represented by the record on the southern Taymyr Peninsula). Also shown are the simulated biome changes  in the surroundings of these sites. In contrast to the simulated biome change, the reconstructions do not show an absolute change in biome coverage. They depict the trend only semi-qualitatively and can be interpreted as change in the similarity of reconstructed pollen taxa composition to a certain biome.

According to the biome reconstructions, cool-mixed forests slightly dominate over steppes and temperate xerophytic shrublands at Lake Daihai in the forest-steppe-desert transition zone during mid-Holocene. After 5.1 cal ka BP, biome composition is characterised by a dominance of steppes and temperate xerophytic shrublands (assigned to desert in the BIOME4-Macro-biome classification). The overall increasing trend in steppe and deserts and decreasing trend in forest is interrupted by periods with substantially raised forest biome similarity around 3.7 cal ka BP and 1.6 cal ka BP. Forest trend is reversing in the last 400 years.

Deciduous taiga is the dominant biome on the Taymyr Peninsula in the taiga-transition zone during the mid-Holocene. The overall change in the biome composition to present day is characterised by a decrease of deciduous taiga and an increase of

the tundra biome. The record, however, shows centennial-scale short-term reversals from this overall trend, e.g. tundra shrubs are strongly increased around 5.7 cal ka BP, 3.8 cal ka BP and around 1.9 cal ka BP. Tundra even dominates at 5.5 cal ka BP and 3.5 cal ka BP. Overall, deciduous taiga stays on a high level until 1.5 cal ka BP. In the last 1500 years, plant composition on the Taymyr Peninsula shows similarity to both, deciduous taiga and tundra biomes. From 0.7 cal ka BP on, tundra shrubs are the dominant biome around the lake.

## 4  Discussion

### 4.1  Differences in biome simulations originating from weaknesses in simulated climate

One aim of this study is the representation of a range of possible biome changes in Asia during the last 6000 years. For this purpose, we forced BIOME4 with simulated climate anomalies from various global climate models. The outcome of this analysis and its reliability depends directly on the capability of the models to simulate the mid-Holocene to pre-industrial
climate change. Biases in the models and differences in the simulated climate between the individual models lead to errors and differences in the simulated biome distributions. Recent studies revealed that climate models have large deficits in representing the present day mean precipitation distribution and seasonal cycle of the Asian monsoon system (Kang et al. 2002; Zhou et al., 2009; Boo et al., 2011) that affect large parts of the area under consideration in this study. Temperature distributions are generally better captured by the models.

A detailed assessment of the performance of the models used in this study with respect to the Asian monsoon precipitation distribution has been conducted in Dallmeyer et al. (2015). In general, the models reveal similar biases regarding the precipitation pattern, but the amplitude diverges. All of the models overestimate the present day annual precipitation in central- and north-eastern China. In the East Asian monsoon transition zone, simulated precipitation exceeds the observations by up to 900 mm/year in the ensemble mean. As we use climate anomalies between the different time-slices and
pre-industrial and add them to observational data, this bias in precipitation should not affect the simulated biome distributions. The models respond very differently to the insolation forcing, revealing large differences in model sensitivity that have an impact on the simulated biome distribution. PLASIM generally shows a rather weak response. The PLASIM simulation ran at coarser resolution than the other simulations. In the pre-industrial control run, the Asian monsoon strength is underestimated by the model (cf. Dallmeyer et al., 2015), thus, the dynamic of the monsoons might not be represented
correctly. The accelerated simulations reveal a rather strong response and exhibit large variability. The latter is at least partly due to the method of averaging the climate forcing data. We took the climatology of 120 calendar years as representative of

the climate at the individual time-slices. These are 120 model years in the non-accelerated simulations, but only 12 years in the accelerated simulations.

In general, the coarse resolution of the models may lead to uncertainties in the biome simulations. Local (sub-grid scale) climate and vegetation changes can not be represented in the simulations, but may be recorded in reconstructions showing mostly the vegetation change in e.g. the catchment area of the lakes. The coarse resolution may, thus, lead to artefacts in the shift of vegetation zones, e.g. if the precipitation increases in one grid-box column at the desert-steppe margin, the steppe also shifts by one grid box column since sub-grid changes are not resolved.

Furthermore, different spatial resolutions lead to different representations of the elevation in the climate models, which may have an impact on the climate change during the Holocene, particularly in high-elevated regions. As we use the anomaly method to calculate the biome distribution, the elevation - or more precisely – the climate gradients resulting from the elevation are preserved from the CRU reference climate. Implicit in this widely used downscaling approach is the assumption that the temperature lapse rate does not change with climate. This is a fair assumption, if not drastically different climate states (e.g. glacial vs. interglacial climate) are considered.

At least partly, the discrepancies in simulated climate may be related to the differences in interactive model components used in the climate model, i.e. some models include dynamic vegetation some do not. Previous modeling studies indicate that land-surface feedbacks with the atmosphere could have enhanced the Holocene precipitation change induced by orbital forcing in the Asian monsoon region (e.g. Wang, 1999; Texier et al., 2000; Diffenbaugh and Sloan, 2002; Li et al., 2009). To test the influence of dynamic vegetation on the simulated Asian climate, sensitivity simulations have to be undertaken. An appropriate set of experiments only exists in the COSMOS-setup (cf. Dallmeyer et al., 2010.). According to these simulations, interactive vegetation has a negligible effect on the mid-Holocene to pre-industrial precipitation change in the desert-steppe-forest transition zone. Vegetation feedbacks contribute to the warmer mid-Holocene climate in the high northern latitudes, but the interactive ocean has a much stronger impact on the climate change. Therefore, the lack of interactive vegetation in KCM and ECHO-G may partly lead to biases in simulated climate change, but we do not expect an effect of this on the general results of this study.

## 4.2 Comparison with pollen-based biome reconstructions

Vegetation reconstructions for Asia are sparse. Particularly in the Monsoon-Westerlies transition zone, i.e. one of the most sensitive Asian regions with respect to climate change, vegetation changes are often poorly documented (Zhao and Yu,

2012). Maps for comparing reconstructed vegetation distribution with model simulations are displayed in the form of biomes for individual sites, but these maps only exist for certain key time-slices and can only give a vague picture of Asian vegetation change. Details in the vegetation change are even more difficult to compare with model results since reconstructed trends – so far - are mostly presented by trends of different taxa at single sites (e.g. Cao et al., 2013) while simulated vegetation in connection with global climate modelling is often described in the form of few plant functional types

(within coupled dynamic vegetation models, e.g. ) or e.g. biomes (within diagnostic vegetation models).

Therefore, we translate representative records of reconstructed plant taxa into biomes that are in accordance with the biome classifications used in BIOME4 (Fig. 11), i.e. the Lake Daihai record (forest-steppe-desert transition zone) and the 13-CH-12 record (taiga-tundra transition zone). The Lake Daihai biome reconstruction, indicating a dominance of cool mixed forests during the mid-Holocene, reflects the expansion of pine forests in the uplands as depicted by the high Pinus-pollen

percentage in the pollen spectra (Xu et al., 2010). However, the low land most probably was covered by steppe vegetation throughout the Holocene. This general trend of a mid-Holocene maximum is known from further sites in north-central China (e.g. Shi and Song, 2003; Jiang et al.,2006; Zhao et al., 2009), while sites on the Tibetan Plateau rather indicate an early-to-early mid-Holocene forest and vegetation maximum (e.g. vanCampo et al., 1996;  Shen et al., 2005, Zhao et al., 2009). At present day, steppe is the dominant biome in the Lake Daihai region, while forests are substantially reduced. Furthermore,

taxa composition has a high resemblance with temperate xerophytic shrublands, probably reflecting an increase in shrubland and desert vegetation in this region. Human interference cannot be excluded at this site (c.f. Xiao et al., 2004), but may have affected vegetation change since the mid-Holocene rather locally.  In Eastern and Central Asia, overall climate is assumed to be the major driver of forest decline (e.g. Wang et al., 2010, Cao et al., 2015, Tian et al. 2016)

The biome simulations presented here support the Lake Daihai reconstruction and are also in line with the general view of

Holocene vegetation change in the forest-steppe-desert transition zone (cf. Sect. 1). The forest fraction is substantially increased (by 10%) and the desert area is reduced (by 21%) during the mid-Holocene. The steppe-to-forest border is shifted approximately 250 km north-westward in the ensemble mean, related to an expansion of the cool/cold forest and taiga biomes into the pre-industrial East Asian steppe. The desert-to-steppe border (including desert and xerophytic shrubland biomes) is even shifted approx. 450 km to the west. However, the models reveal strong discrepancies among each other with

respect to the vegetation distribution and temporal change. Regarding the vegetation change around Lake Daihai (in the model: 110-114°E, 38-42°N), the models mostly agree with the biome reconstruction on the long term, showing a decrease of cool/cold forest and an increase of grassland and shrublands since the mid-Holocene in the ensemble mean (cf. Fig. 11). They even reveal short-time recoverage of the forest biome at certain phases, e.g. around 5k, 2k and 1k. The timing of these

phases deviates from the reconstruction, but as mentioned before, the spread in the simulated vegetation change between the
different climate models is large. Furthermore, the catchment area of Lake Daihai is not exactly definable and does most
probably not coincide with the region chosen in the model. The models reveal regionally strong differences in the temporal
forest and desert change within the forest-steppe-desert transition zone that may not be captured by the reconstructions. And
last but not least, short-term (multi-centennial) variability may not be resolved in the biome simulations that have been run
with coarse temporal resolution (time steps of 500 years).

The reconstructed Holocene increase of tundra and decrease of taiga shown in the Taymyr Peninsula record from north-
central Siberia is in agreement with other Holocene records from northern Siberian treeline areas (e.g. McDonald et al.,2000,
Herzschuh et al.,2013). It reflects the southward retreat of larches that coincide with the southward shift in the boreal treeline
since the mid-Holocene due to cooler climate at 0k.

While the deciduous taiga biome reveals a clear dominance during mid-Holocene around Lake 13-CH-12, tundra covers the
landscape at present day. This is not in line with the biome simulations for pre-industrial (based on the CRU-climate dataset),
showing deciduous taiga and boreal woodlands in the region around the lake. This difference probably results from the non-
equilibrium of the regional vegetation with the fast warming climate in the Arctic region. Thus, tree coverage is
overestimated by the model, since temperature is the limiting factor for tree growth in this region. Furthermore, only few
meteorological stations exist in northern Siberia. This fact may lead to larger uncertainties in the CRU-dataset for northern
Siberia compared to other regions.

Due to the overestimation of tree coverage around the lake by the biome model, we chose a corresponding region at the
taiga-tundra border for comparison with the reconstructed biome trends (106-110°E,69-73°N).

In general, the biome simulations are in line with the general view of reconstructed Holocene vegetation change at the boreal
treeline and the record from the Taymyr Peninsula. The taiga and boreal woodlands were expanded northward during mid-
Holocene, shifting the boreal treeline by approx. 4° to the north. The simulated biome change at the site is large. Vegetation
switches from coverage by 100% of taiga during the mid-Holocene to coverage by 80.5% tundra for pre-industrial. Similar
to the reconstructions, the simulations also reveal periods with substantially increased tundra, i.e. at 4k and at 1k.

## 5  Summary

To get a systematic overview on the Holocene vegetation change in Asia, the terrestrial biosphere model BIOME4 has been
forced with climate output of five different transient simulations performed with coupled atmosphere-ocean(-vegetation)

models. The differences in biome distribution between the mid-Holocene and pre-industrial (potential vegetation) as well as the temporal change in Holocene vegetation have been analysed.

The model results reveal only weak biome changes in Asia during the last 6000 years. Main changes concentrate on the high northern latitudes and the transition zone from the moist East Asian summer monsoon to the dry Westerly winds. In the high northern latitudes the northern treeline shifts northward due to an expansion of taiga during the mid-Holocene. The vegetation in the Monsoon-Westerlies transition area consists of desert, grassland and forest belts, whose margins shift in time. In the majority of cases, the models agree regarding the pattern of biome shift, but the amplitude of change strongly depends on the climate forcing, i.e. the different climate models.

According to BIOME4, the vegetation changes in the forest-steppe-desert transition zone (95-125°E, 32-52°N) are mainly influenced by changes in precipitation. The higher moisture levels in the mid-Holocene lead to a decreased desert fraction by approx. 20% and a westward shift of the desert-steppe boundary by approx. 500 km in the ensemble mean, ranging from 100-1000 km in the different simulations. The expansion of desert during the Holocene back to modern distribution is not uniform and varies spatially. Depending on the climate forcing and the longitudinal position, the general trend of declining vegetation is interspersed with periods of substantially increased or decreased desert fraction.

Forests are increased by approx. 8% during the mid-Holocene, penetrating into the steppe and desert regions from two sides. On the one hand, the increased precipitation at 6k leads to a westward shifted forest-steppe boundary in East China by 2°, ranging from 0°-4° in the different simulations. On the other hand, the forest fraction is increased on the Altai Mountains and the Tibetan Plateau replacing mainly tundra vegetation. In between (103-110°E), the area covered by forest biomes is reduced at 6k. Averaged over the entire region, the maximum forest extent is reached at 5k and 2.5k (12%) in the ensemble mean. Generally, the biome simulations are in line with time-slice reconstruction for the mid-Holocene and with the reconstructed biome trend at Lake Daihai.

Biome shifts in the north-central Siberian taiga-tundra transition zone (66-120°E, 60-80°N) are mainly related to temperature changes. The annual mean temperature rise and the associated increase in growing degree days results in an expansion of taiga during the mid-Holocene. The tundra fraction in the high northern latitudes is decreased by 64% in the ensemble mean, ranging from 48%-75% in the different simulations. Accordingly, the taiga border is shifted northward by 150-600 km in the individual simulations and by approx. 400 km in the ensemble mean. From 6k to 4k, the taiga retreats relatively fast, afterwards slowly and gradually. Some individual simulations reveal high temporal variability of tundra fraction. The biome simulations agree with vegetation reconstructions, but the taiga coverage is overestimated by the model, probably due to a non-equilibrium of the northern Siberian vegetation to the fast warming climate. One reason explaining the non-linear retreat

of the Taiga relates to the non-linear decline in northern hemisphere summer insolation in the course of the mid-to-late Holocene. This non-linear decline might further have been amplified by internal climate feedbacks, for instance related to the expansion and persistence of Arctic sea ice during summer.

Besides the general conclusions based on the model results one also should account for uncertainties that are included in the present results: Most models used have a quite coarse horizontal resolution of several hundreds of kilometres – therefore

changes in the longitudinal and latitudinal spatial extend of certain biomes should show at least this order of magnitude to allow a robust conclusion. More pollen records are needed to evaluate the simulated results, particularly in the desert-steppe transition zone.

Second and related to the last point is the already mentioned partly unrealistic representation and simulation of precipitation in the models. To overcome this problem one needs to carry out a regionalization of the GCM output by means of statistical

and/or dynamical downscaling to yield a better basis for the BIOME4 model in terms of hydrological changes or even better, run global climate models with higher horizontal resolution. This methodological step is however beyond the scope of this study; results that are based on thermal changes in the course of the Holocene can be expected to be more realistically simulated by the GCMs.

Our way to account for the obvious shortcomings in the simulation of precipitation by the GCMs was a bias correction based

on our anomaly approach using the CRU TS3.10 data based on observed meteorological data.

**6. Conclusion**

This study assess the Holocene vegetation change in Eastern Asia. While the spatial distribution of biomes has rather been stable in most of the investigated areas since the mid-Holocene, this study identifies two main regions with strong vegetation changes, i.e. the desert-steppe-forest transition zone related to the East Asian monsoon margin and the taiga-tundra transition

zone in northern Siberia. Both represent transition zones between forested and non-forested biomes. Particularly the East Asian monsoon margin has hitherto been poorly documented in reconstruction. The strong spread of the simulated Holocene biome distributions shows that the vegetation in these regions is very sensitive to variations in climate, i.e. to precipitation changes in Central Asia and to temperature change in northern Siberia. Accordingly, these areas can be expected to react strongest to ongoing climate change.

## Acknowledgements

This research is part of the priority program INTERDYNAMIK funded by the German Research Foundation (DFG). It also contributes to the project PalMod, funded by the Federal Ministry of Education and Science (BMBF). We thank J. Bader (MPI-M) and three anonymous reviewer for constructive comments that helped to improve the paper. The modified Biome4 model, input data and scripts used in the analysis and other supplementary information that may be useful in reproducing the authors' work are archived by the Max Planck Institute for Meteorology and can be obtained by contacting publications@mpimet.mpg.de

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

**Appendix**

**A Additional figures showing the modifications of the BIOME4 model**

Appendix A provides three figures which highlight the differences in the BIOME-4 model as a result of our modifications.

**B Bioclimatic changes**

Since the simulated vegetation distributions strongly depend on the given bioclimatic conditions, we additionally identify marked changes in bioclimate between mid-Holocene and pre-industrial. Figure B.a shows the modern pattern of different bioclimate variables that determine the biome distribution in Asia based on the CRU TS3.10 dataset (cf. Tab. 1). To express

the moisture limitation in the semi-arid and arid regions of Asia, the annual mean precipitation and the Taylor-Priestley coefficient (alpha) are additionally provided. Figure B.b presents the ensemble mean difference in these bioclimatic variables between mid-Holocene (6k) and modern climate (0k). During mid-Holocene, the annual mean near-surface air temperature is increased by up to 1.2 K in the high northern latitudes and subtropics and decreased by up to 1.9 K in the tropics, attenuating the zonal temperature contrast. This temperature change is accompanied by an increase in growing degree days (GDD5)

north of 30°N that is particularly pronounced in the region around the Hami and Turpan depression (up to 240°C) and in western Siberia (up to 220°C). South of 30°N, growing degree days are reduced by up to 650°C. The mean near-surface temperature during the warmest month is higher in the entire Asian region aside from India and parts of the Indochina peninsula. The maximum warming occurs on the Tian Shan mountain range with a temperature increase by up to 3.2 K in the ensemble mean. The temperature of the coldest month ($T_{cm}$) is strongly increased in East Siberia by up to 1.8 K during mid-

Holocene. In the other regions, $T_{cm}$ is markedly lower at 6k with anomalies reaching up to 2.7 K in India, corresponding to the decreased winter insolation during mid-Holocene.

The annual mean precipitation change is particularly pronounced in northern India, where the enhancement of the Indian summer monsoon during mid-Holocene leads to an increase in precipitation by up to 300 mm/year. Precipitation is also enhanced in North Eastern China and Inner Mongolia (up to 120 mm/year), probably resulting from a strengthening and

further inland penetration of the East Asian monsoon, and in East Siberia (up to 80 mm/year). In these areas, the increase in precipitation leads to a rise in moisture availability, i.e. the Taylor-Priestley coefficient is increased by up to 0.22. In contrast, parts of West Siberia and East China experience slightly less precipitation and decreased moisture availability during mid-Holocene.

## C  Transient biome changes in the different simulations

Fig. C1 shows the simulated meridional mean change in desert coverage in the desert-steppe-forest transition area since the mid-Holocene for each simulation. Details not discussed in Chapter 3 are presented here: In COSMOS, the desert is confined to the area west of approximately 107°E at 6k. The desert fraction is particularly decreased (by more than 20% of the modern fraction) in the region between 107°E and 113°E during the mid-Holocene. This biome distribution stays relatively constant until 4.5k. Afterwards, the desert border gradually moves eastward to its pre-industrial position accompanying a general

increase in the desert fraction that first starts in the western part (around 108°E at ca. 4.5k), and latest in the eastern part of the transition zone (starting at 3k around 111°E and at 2k around 113°E). However, the expansion of the desert is longitudinally not uniform: Areas around 102°E and 113°E  show a relatively high degree of vegetation cover until 0.5k, periods with increased desert fraction that partly even exceeds the pre-industrial fraction interrupt the periods showing higher vegetation cover between 108°E and 110°E.

ECHO-G shows a much weaker response. The desert-steppe border is shifted westward by approx. 3° at the mid-Holocene and reaches its modern position around 1000 yrs before present (BP). The desert fraction in the transition zone is decreased by about 10-20% between 6k and 2k, particularly in the region around 110-113°E. Similar to COSMOS the period of enhanced vegetation is interrupted by periods with higher desert fraction than pre-industrial. During the last 1000 years BIOME4 suggests less vegetation than today in parts of the transition zone (ca. 108°-111°E).

The simulation based on PLASIM indicates a very different picture of Holocene biome changes in the desert-steppe transition zone. Large parts of the region (ca. 98°E-109°E) reveal a higher percentage of desert area during most periods of the last 6000 years. However, directly in the region around the desert-steppe border (ca. 110-115°E) vegetation is increased by up to 3.5% compared to the modern distribution. The steppe extent is relatively constant.

In contrast, the change in biome distribution based on the two accelerated simulations, i.e. COSMOSacc and KCM, show a

strong variability. The generally increased vegetation cover during the Holocene compared to pre-industrial is interrupted by several periods with substantially reduced vegetation. The most prominent event occurs around 1.5k. Similar as for ECHO-G, KCM suggests less vegetation in parts of the transition zone during the last 500 years. BIOME4 shows the strongest response using COSMOSacc as climate forcing. The desert fraction is decreased by more than 20% in large parts of the region from 6k to 2k and the mean desert border is shifted to 104°E at 6k.

Fig. C2 shows the meridional mean change in forest fraction in the desert-steppe-forest transition zone for the individual simulations. Directly west of the modern mean forest border (120°E),  COSMOS and KCM reveal the strongest change with an increase of forest fraction by more than 20% and a westward shift in the forest-steppe-border by approx. 3°. PLASIM suggests a weak change in forest coverage since the mid-Holocene. The forest fraction is increased between 113°E and 118°E from 6k to 1k, but decreased westward of 118°E by more than 10% during the entire period, actually resulting in an

eastward shift of the mean forest-steppe-border.

60

ECHO-G and PLASIM suggest slightly less forest area compared to the modern distribution in nearly the entire region during the last 1000 years, while KCM and COSMOSacc still show enhanced forest cover but with a strong decreasing trend during this period. The simulations ECHO-G, KCM and COSMOSacc are characterised by strong variability revealing several periods with substantially reduced forest fraction (e.g. 4k and 1.5k in ECHO-G, 3.5k in COSMOSacc, 3k and 1.5k in KCM). The maximum forest extent in the simulations based on the accelerated climate models is not reached during the mid-Holocene but at 5k and 4k in COSMOSacc and 4.5k, 2k and 1k in KCM, showing an increased forest fraction by more than 30% and 20% during these periods, respectively.

Fig.C3 shows the zonal mean change in tundra fraction in the north-central Siberian taiga-tundra-transition zone since the mid-Holocene for all simulations.

According to COSMOS, the tundra fraction is decreased by more than 25% in large parts of the north-central Siberian taiga-tundra transition zone during mid-Holocene. Directly at the mean taiga border, the tundra area is reduced by more than 50%. This leads to a shift of the mean taiga border to 73.5°N at 6k compared to 67.5°N at pre-industrial. COSMOS reveals a period with substantially enhanced tundra fraction around 4k (up to 25%), resulting in a southward retreat of the taiga border to 68°N. The last 1000 years are characterised by slightly less taiga compared to pre-industrial. ECHO-G shows a similar signal, but reveals an uninterrupted and rather gradual retreat of the taiga during the Holocene.

In PLASIM, the overall response to the Holocene climate forcing is the weakest. The mean taiga border is shifted northward by only 1.5° at 6k. The modern vegetation distribution is already established at 1k. In contrast, COSMOSacc shows the strongest signal, suggesting a reduction of the tundra fraction by more than 50% in large parts of the region at mid-Holocene. The mean taiga border is shifted to 73.5°N at 6k and stays at this position until 5k. Afterwards, the taiga forest rapidly retreats southward of 69.5°N. The variability in tundra fraction is relatively high, revealing periods with substantially decreased tundra area compared to other time-slices at 4k and 2.5k. During the latter period, tundra is reduced by approx. 67%, averaged over the entire region. At 1.5k, COSMOSacc suggests a period with strongly increased tundra fraction north of 70°N by up to 25%. In KCM, the mean taiga border is located at 71°N during mid-Holocene and the tundra fraction is reduced by more than 25% in large parts of the region. The taiga moves gradually back to the modern position, only interrupted by a period with expanded taiga (up to 71°N) at 4.5k and a period with increased tundra at 1k.

| PFT | Tcm min | Tcm max | Tmin min | Tmin max | GDD5 min | GDD0 min | Twm min | Twm max |
|---|---|---|---|---|---|---|---|---|
| Trop. evegreen trees | - | - | 0.0 | - | - | - | 10 | - |
| Trop. deciduous trees | - | - | 0.0 | - | - | - | 10 | - |
| Temp. broadleaved trees | 2 | - | -8.0 | 5 | 1200 (1500) | - | 10 | - |
| Temp. deciduous trees | -15 | | - | -8 | 1200 | - | - | - |
| cool conifer | -2 | - | - | 10 | 900 | - | 10 | - |
| Bor. evergreen trees | -32.5 | -2 | - | - | 350 (500) | - | - | - |
| Bor. deciduous trees | - | 5 | - | -10 | 350 (500) | - | | - |
| C3/C4 temp.grass | - | - | - | 0 | 550 | - | - | - |
| C4 tropical grass | - | - | -3 | - | - | - | 10 | - |
| woody desert | - | - | -45 | - | 500 | - | 10 | - |
| tundra shrub | - | - | - | - | - | 50 | - | 15 |
| cold herbaceous | - | - | - | - | - | 50 | - | 15 |
| lichen/forb | - | - | - | - | - | | - | 15 |


**Table 1: Bioclimatic limits used in BIOME4 to express the climatic tolerance of different plant functional types (PFT), i.e. mean temperature of the coldest month (Tcm), absolute minimum temperature (Tmin), mean temperature of the warmest month (Twm) and growing degree days based on 0°C and 5°C (GDD0, GDD5). The limits in the brackets represent the climatic tolerance on mountains (altitudinal vegetation constraints) to which the model switches in grid-boxes with orography exceeding 2000 m.**


| No. | Biome | Mega-Biomes | Macro-Biomes |
|---|---|---|---|
| 1 | Tropical evergreen forest | tropical forest | forest |
| 2 | Tropical semi-deciduous forest | | |
| 3 | Tropical deciduous forest/woodland | | |
| 4 | Temperate deciduous forest | temperate forest | |
| 5 | Temperate conifer forest | | |
| 6 | Warm mixed forest | warm forest | |
| 7 | Cool mixed forest | cool/cold forest | |
| 8 | Cool conifer forest | | |
| 9 | Cold mixed forest | | |
| 10 | Evergreen taiga/montane forest | evergreen taiga | |
| 11 | Deciduous taiga/montane forest | deciduous taiga, boreal woodland | |
| 17 | Open conifer woodland | | |
| 18 | Boreal parkland | | |
| 12 | Tropical savannah | savannah/ woodland | |
| 13 | Temperate broadleaved savannah | | |
| 14 | Temperate sclerophyll woodland | | |
| 19 | Tropical grassland | Grassland | steppe |
| 20 | Temperate grassland | | |
| 15 | Temperate xerophytic shrubland | Shrubland | desert |
| 16 | Tropical xerophytic shrubland | | |
| 21 | Desert | Desert | |
| 22 | Steppe tundra | Tundra | tundra |
| 23 | Shrub tundra | | |
| 24 | Dwarf shrub tundra | | |
| 25 | Prostrate shrub tundra | | |
| 26 | Cushion forb lichen moss tundra | | |
| 27 | Barren | barren/ice | |
| 28 | Land ice | | |

**Table 2: Biomes calculated in BIOME4 and their classification into mega-biomes and macro-biome that are considered in this study (cf. Fig. 2)**

| Abbreviation | Model | Reference | Res. | dyn.veg. | forcing | acc. |
|---|---|---|---|---|---|---|
| COSMOS | ECHAM5/JSBACH-MPIOM | Fischer and Jungclaus (2011) | T31L19 | yes | orbit | no |
| ECHO-G | ECHAM4-HOPE-G | Wagner et al. (2007) | T31L19 | no | orbit | no |
| PLASIM | PlaSim-LSGocean-Simba | Haberkorn (2013) | T21L10 | yes | orbit + GHG | no |
| COSMOSacc | ECHAM5/JSBACH-MPIOM | Varma et al. (2012) Pfeiffer and Lohmann (2013) | T31L19 | yes | orbit | yes |
| KCM | ECHAM5-NEMO | Jin et al. (2014) | T31L19 | no | orbit | yes |


**Table 3: Overview on all climate simulations used as forcing for BIOME4.**

|         | 0k | 6k | | | | | |
|---------|----|--------|--------|--------|-----------|-----|---------|
|         |    | COSMOS | ECHO-G | PLASIM | COSMOSacc | KCM | ENSMEAN |
| forest  | 45 | 50 | 49 | 43 | 54 | 49 | 49 |
| steppe  | 18 | 23 | 20 | 20 | 23 | 23 | 22 |
| desert  | 26 | 19 | 23 | 23 | 15 | 20 | 21 |
| tundra  | 10 | 7  | 7  | 9  | 8  | 8  | 8  |

**Table 4: Mean coverage of Macro-Biomes [absolute %] in the forest-steppe-desert transition zone (95°-125°E, 32°-52°N) according to the reference simulation (0k) and the different simulations using mid-Holocene climate .**

| border | 0k | 6k | | | | | |
|---|---|---|---|---|---|---|---|
| | | COSMOS | ECHO-G | PLASIM | COSMOSacc | KCM | ENSMEAN |
| desert-steppe | 113°E | 107°E | 110°E | 112°E | 104°E | 107°E | 108°E |
| forest-steppe | 120°E | 117°E | 119°E | 124°E | 119°E | 117.5°E | 118°E |
| taiga-tundra | 67.5°N | 73.5°N | 73.5°N | 69°N | 73.5°N | 71°N | 71.5°N-73°N |


**Table 5: Summary of the mean positions of the desert-steppe (20%-isoline of desert fraction), forest-steppe (80%-isoline of wood fraction) and taiga-tundra border (20%-isoline of tundra fraction) in the different simulations during pre-industrial and mid-Holocene.**


| | 0k | 6k | | | | | |
|---|---|---|---|---|---|---|---|
| | | COSMOS | ECHO-G | PLASIM | COSMOSacc | KCM | ENSMEAN |
| forest | 73 | 91 | 93 | 86 | 93 | 89 | 90 |
| tundra | 27 | 9 | 7 | 14 | 7 | 10 | 9 |

**Table 6: Mean coverage of Macro-Biomes [absolute %] in the north-central Siberian taiga-tundra transition zone (66°-120°E, 60°-80°N),according to the reference simulation (0k) and the different simulations using mid-Holocene climate .**


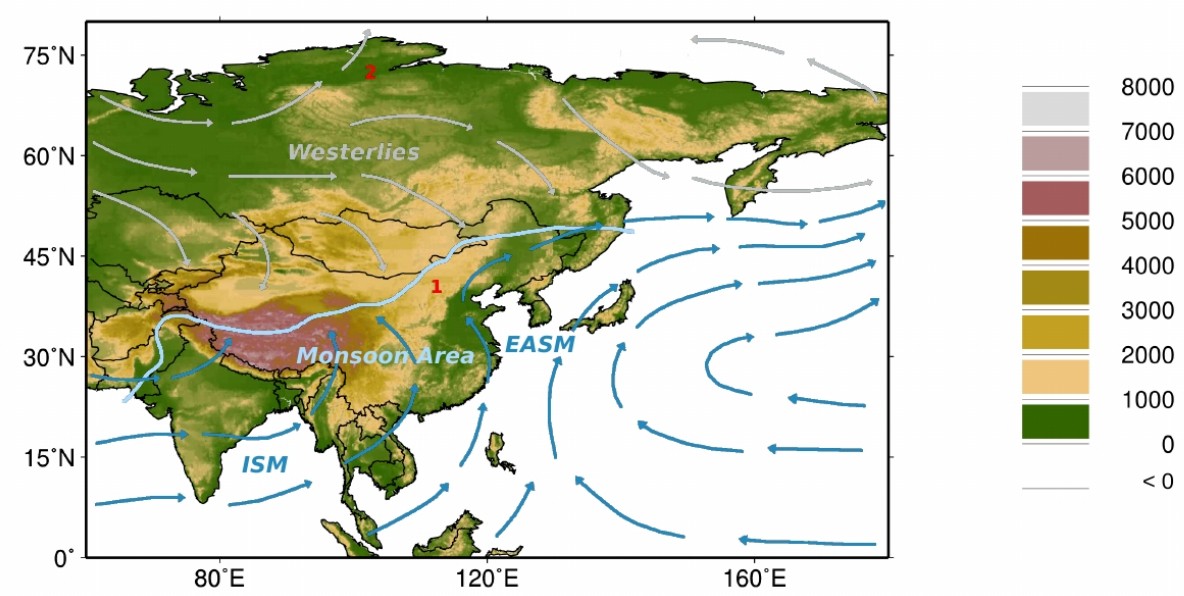

**Figure 1: Orography (shaded, [m]), based on ETOPO5 (Data Announcement 88-MGG-02: Digital relief of the Surface of the Earth. NOAA, National Geophysical Data Center, Boulder, Colorado, 1988) and sketch of mean present-day summer circulation in 850hPa (based on ERA reanalysis data, Uppala et al., 2005) displaying the three major circulation systems (vectors) affecting Asia,**
**i.e. the Indian summer monsoon (ISM, darkblue), the East Asian summer monsoon (EASM, darkblue) and the Westerlies (grey). Shown is also the present-day extend of the Asian summer monsoon region (lightblue) based on the observed mean 2mm/day-summer-isohyet (taken from GPCP data, Adler et al., 2003) . Red numbers mark the location of the pollen-sites, i.e. 1: Lake Daihai (40.5°N, 112.5°E, Xu et al., 2009) and 2: small lake on southern Taymyr peninsula (13-CH-12, 72.4°N, 102.29°E, Klemm et al., 2015).**


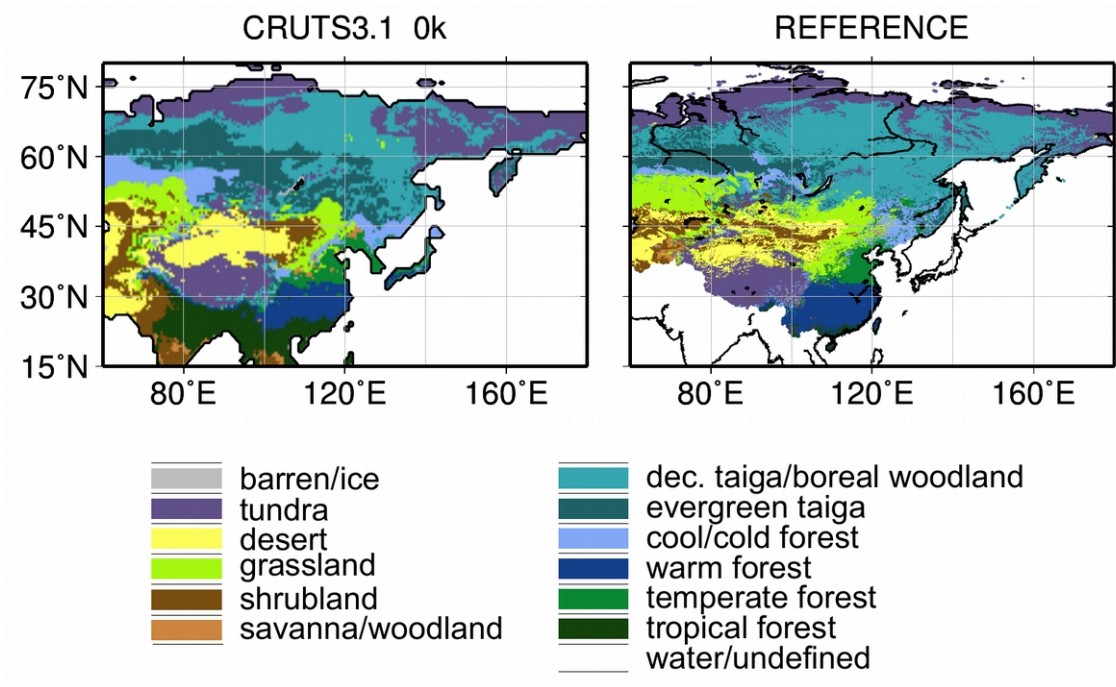

**Figure 2: Simulated biome distribution based on the University of East Anglia Climatic Research Unit Time Series 3.10 (CRU TS3.10, University of East Anglia, 2008, Harris et al., 2012) in comparison to the reference dataset (modified from Hou, 2010; Saandar and Sugita, 2004; and Stone and Schlesinger, 2003)**

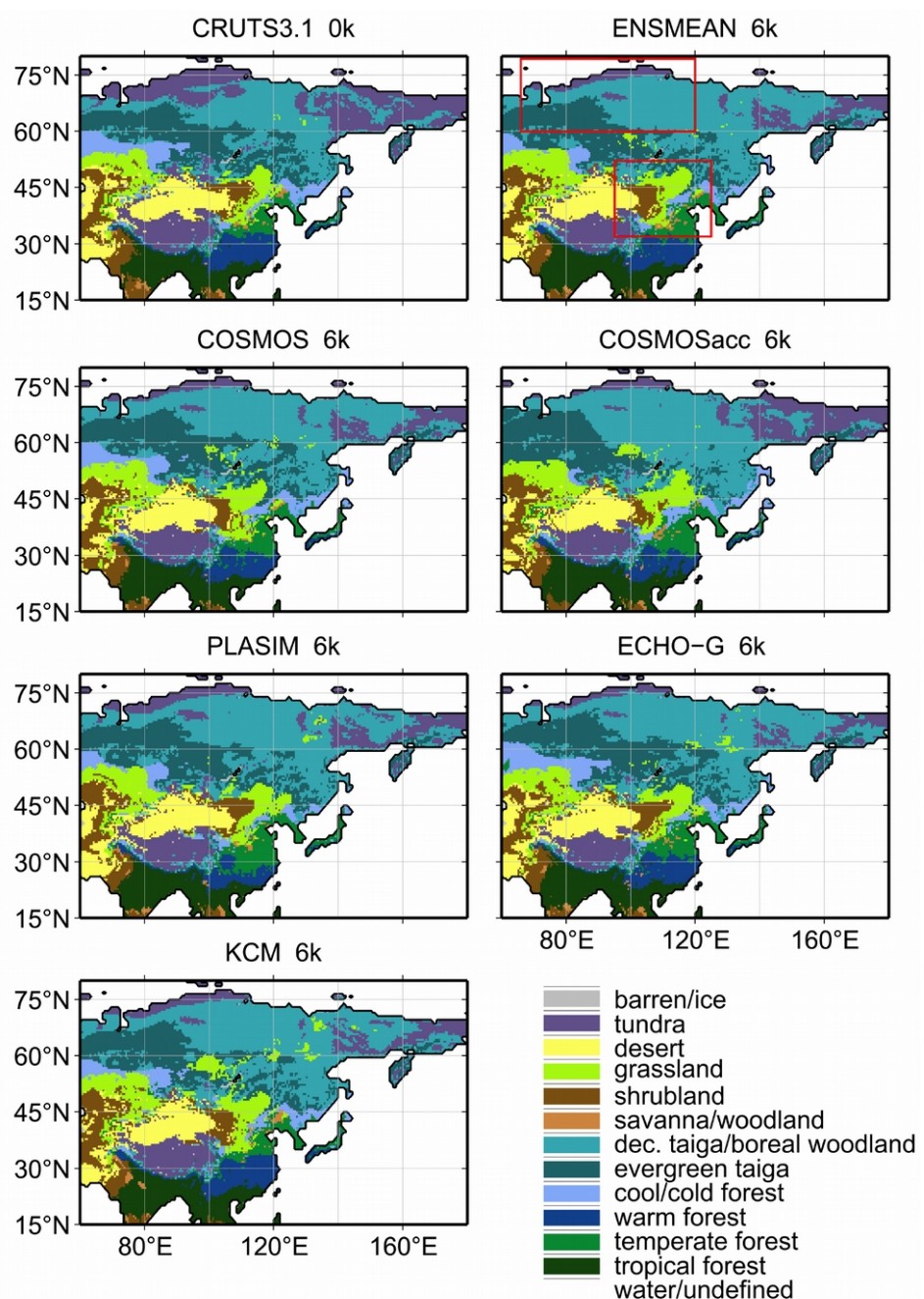

**Figure 3: Simulated biome distributions for the modern climate (CRU TS3.10 0k) and for the mid-Holocene time-slice (6k) based on the ensemble mean climate (ENSMEAN) and the different climate model simulations. In the plot of the ensemble mean, the two key regions of biome changes are marked (red boxes), i.e. the taiga-tundra transition zone in north-central Siberia (66°-120°E,60°-80°N) and the forest-steppe-desert transition zone in north-central China (95°-125°E,32°-52°N).**


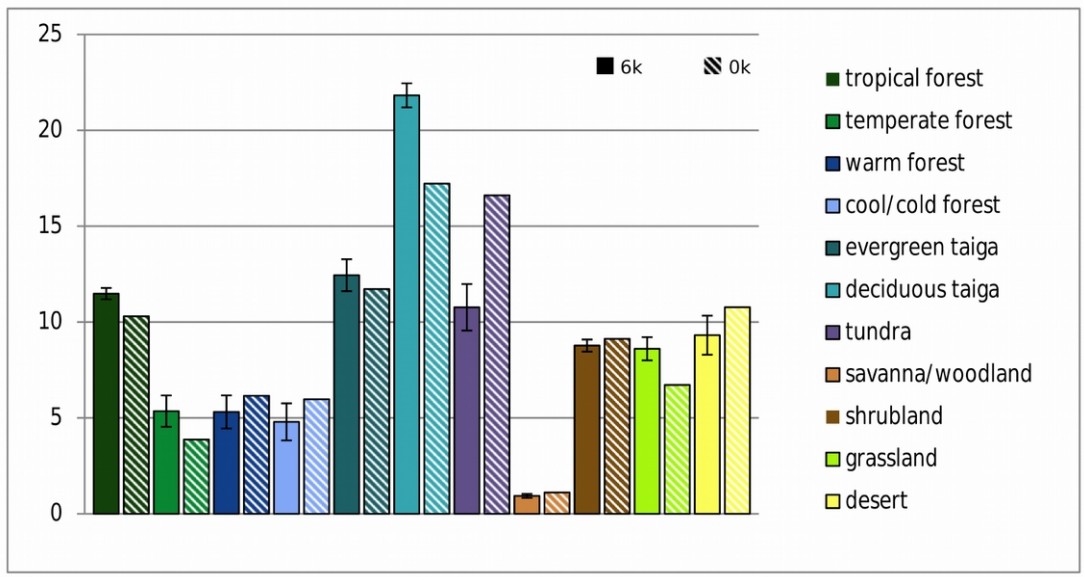

**Figure 4: Simulated percentages of land area [%] for each biome at pre-industrial (hatched) based on CRU TS3.10 and mid-Holocene (fully shaded) averaged over the entire region considered here (60-180°E, 15-80°N) and averaged over all simulations, including the ensemble standard deviation (error-bars).**

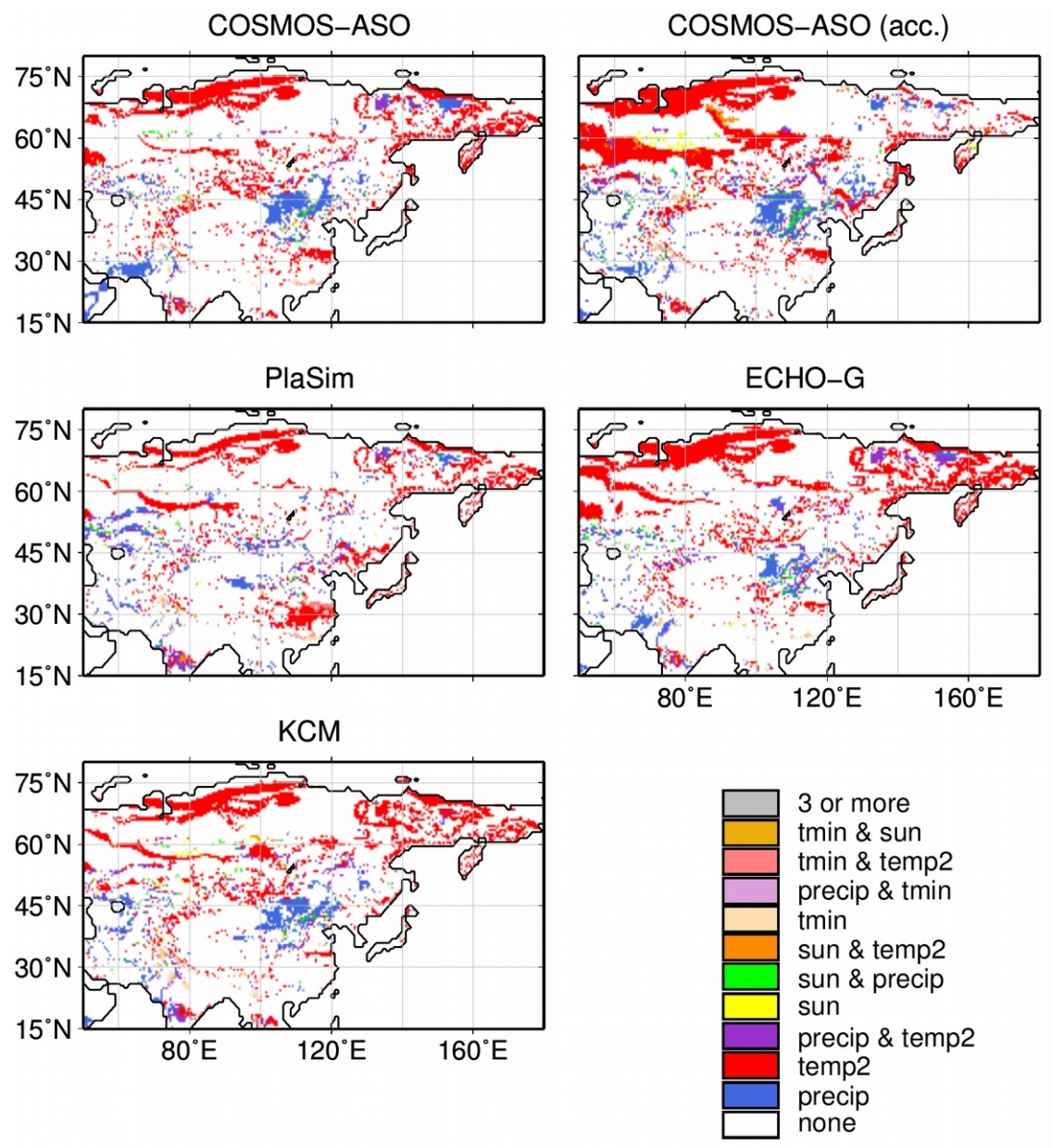


**Figure 5: Factors causing a change in biome distribution as simulated by BIOME4 forced with climate data from different climate model simulations, i.e. precipitation (precip), near surface air temperature (temp2), sunshine/cloudiness (sun) or absolute minimum temperature (tmin).**

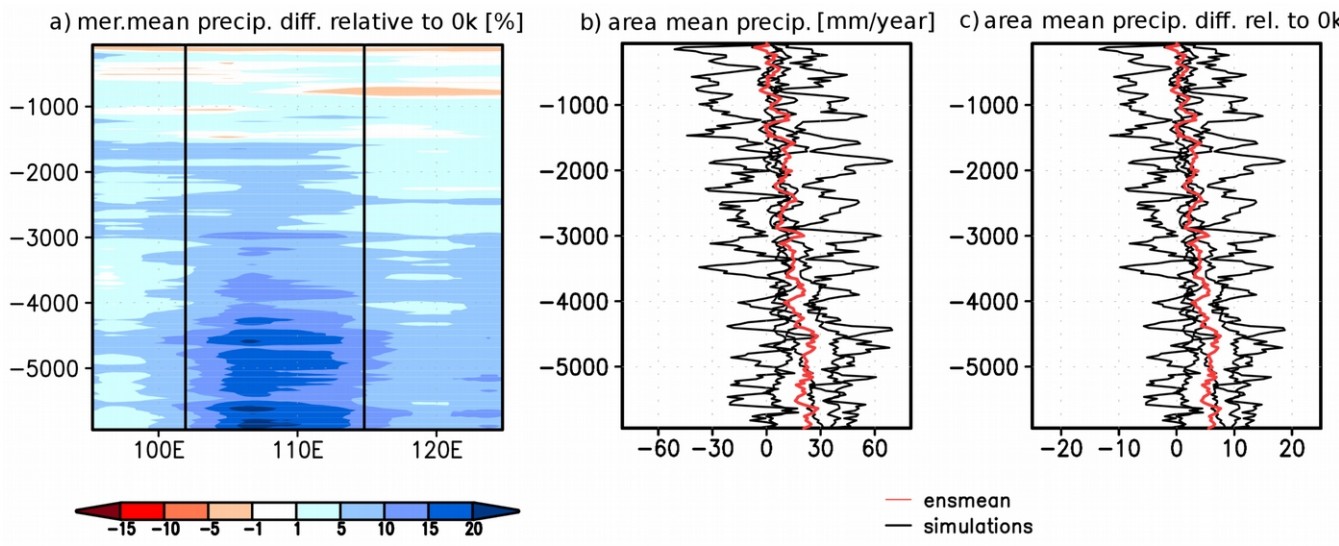

**Figure 6: Mid-Holocene to pre-industrial precipitation change in the desert-steppe-forest transition zone (95°-125°E, 32°-52°N, land only). a)** Hovmoeller diagram showing the ensemble mean of the meridional mean (32°-52°N) relative difference in annual precipitation [%] compared to pre-industrial, the black lines mark the regions with strongest signal; **b)** area mean precipitation [mm/year] averaged over all simulations (red) and for each single simulation (black); and **c)** area mean relative change in precipitation compared to pre-industrial [%].

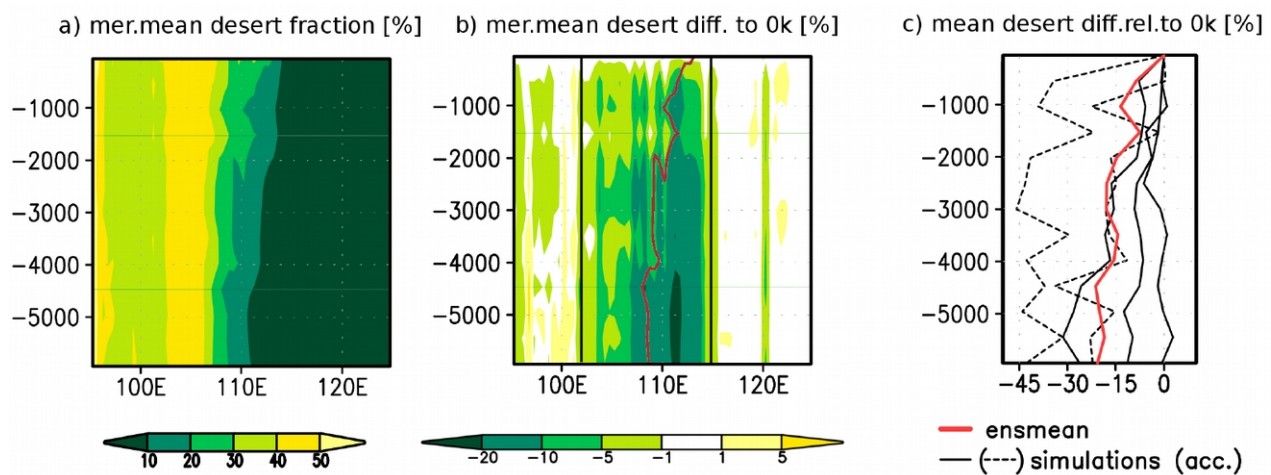

**Figure 7: Mid-Holocene to pre-industrial desert biome change in the desert-steppe-forest transition zone (95°-125°E, 32°-52°N, land only). a)** Hovmoeller diagram showing the ensemble mean of the meridional mean (32°-52°N) desert fraction [% fraction of area], **b)** Hovmoeller diagram showing the absolute change in meridional mean desert fraction [%] compared to pre-industrial as ensemble mean. The redish line marks the steppe border (here ad hoc defined as the isoline of 20% desert fraction) the black lines mark the regions with strongest precipitation change (c.f. Fig.6), **c)** area mean relative change in desert fraction compared to pre-industrial averaged over all simulations (red) and for each single simulation (black). The dashed line displays the accelerated simulations.


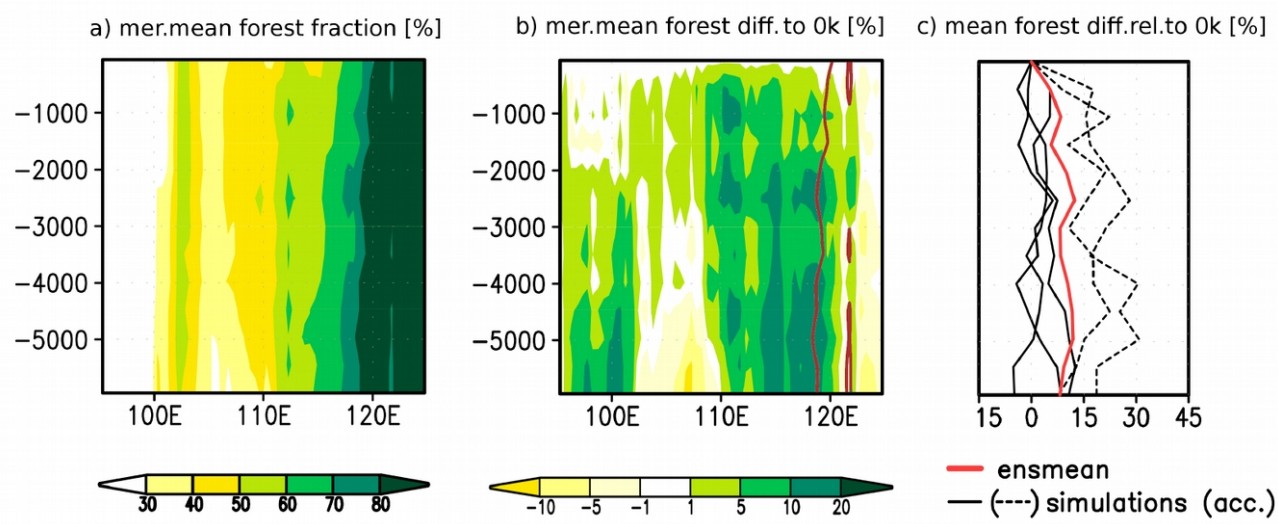

**Figure 8: Same as Fig.7, but for forest fraction. b) redish line displays the steppe-forest margin (here ad hoc defined as isoline of 80% forest fraction).**

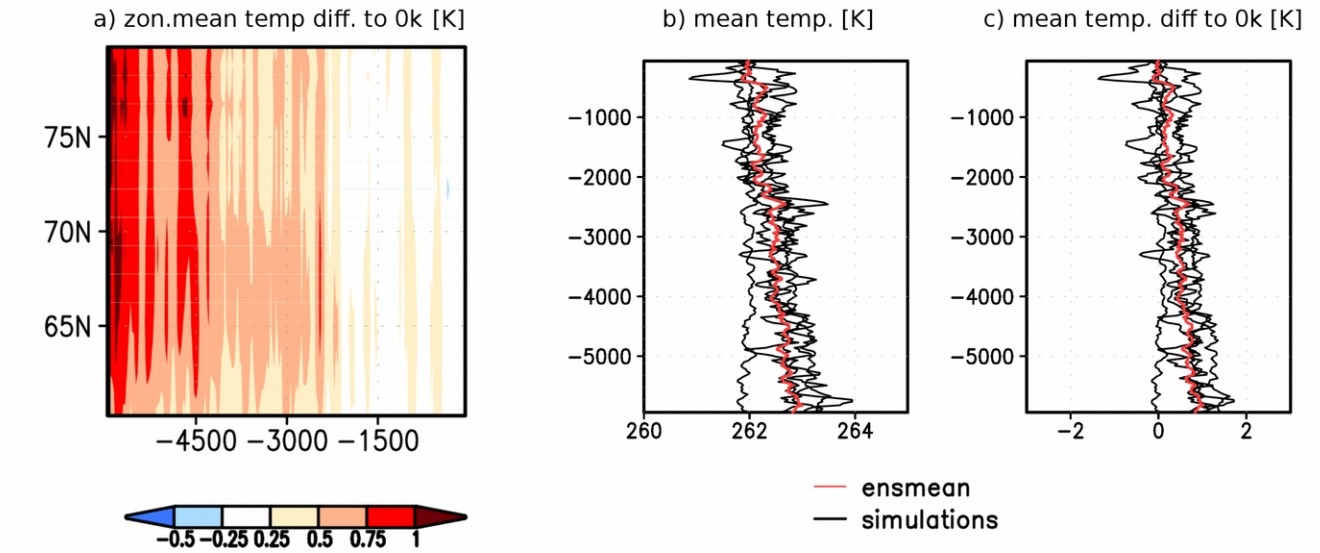

**Figure 9: Mid-Holocene to pre-industrial 2m air-temperature change in the north-central Siberian taiga-tundra margin (66°-120°E, 60°-80°N, land only). a) Hovmoeller diagram showing the ensemble mean of the zonal mean absolute difference in annual mean near-surface air temperature [K] compared to pre-industrial, b) area mean annual mean 2m air-temperature [K] averaged over all simulation (red) and for the individual simulations (black), and c) area mean absolute difference in annual mean near-surface air temperature compared to pre-industrial averaged over all simulation (red) and for each simulation (black).**

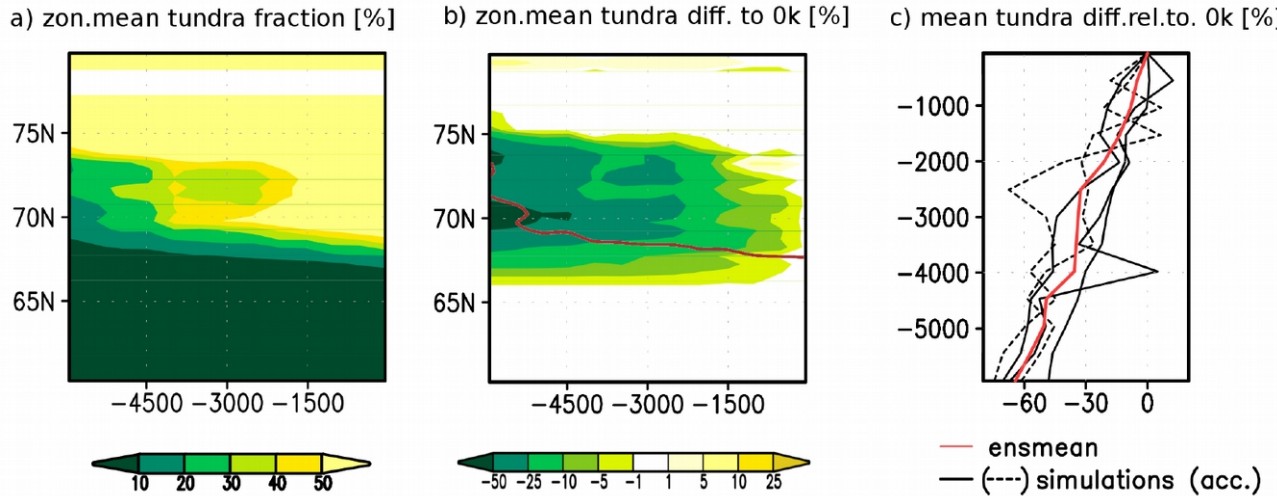

**Figure 10: Mid-Holocene to pre-industrial tundra fraction change in the north-central Siberian taiga-tundra margin (66°-120°E, 60°-80°N, land only). a) Hovmoeller diagram showing the ensemble mean of the zonal mean tundra fraction [% fraction of area], b) same as a) but for the absolute change in zonal mean tundra fraction [%] compared to pre-industrial as ensemble mean. The redish line marks the taiga border (here ad hoc defined as the isoline of 20% tundra fraction); c) area mean relative change in tundra fraction compared to pre-industrial averaged over all simulation (red) and for each simulation (black, dashed: accelerated simulations).**

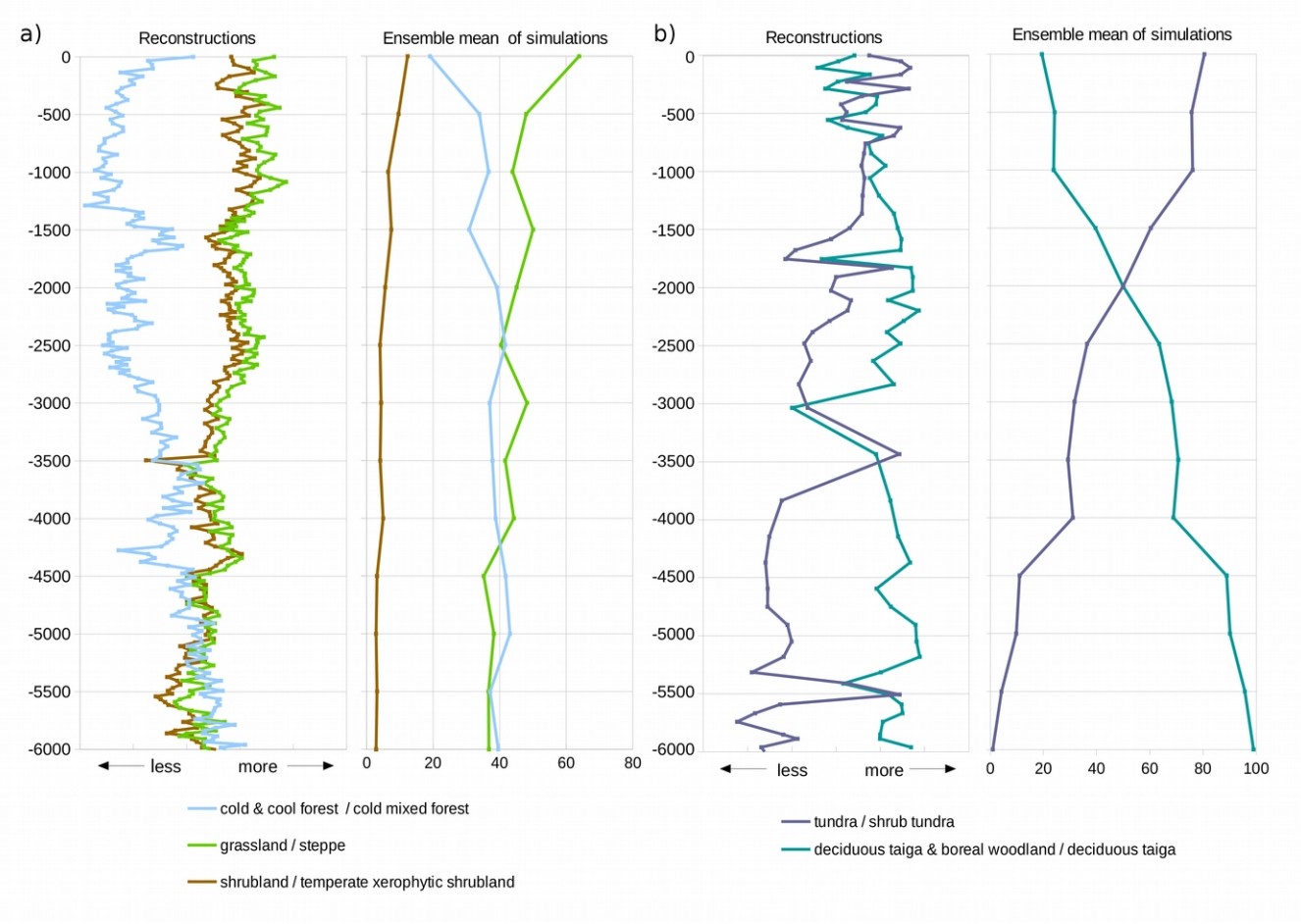

Figure 11: Reconstructed biome change at two representative key-sites, i.e.: a) Lake Daihai (40.5°N,112.5°E, Xu et al., 2009) for the forest-steppe-desert transition zone, and b) a small lake on the southern Taymyr peninsula (13-CH-12, 72.4°N, 102.29°E, Klemm et al., 2015) for the taiga-tundra margin. The reconstructions are given in arbitrary units and show the dominant biome at a certain time and the trend qualitatively (i.e. less or more dominant than other biomes). Shown are also the simulated biome change in the surrounding of both sites, i.e. 110°E-114°E, 38°N-42°N for Lake Daihai and 106°E-110°E, 69-73°N for the southern Taymyr record. The latter region had to be adjusted to the modern taiga-tundra margin that is shifted northward in the modern biome simulation due to an overestimation of forest cover in the boreal region. For comparison, the nearest grid-box around the lake showing tundra vegetation in modern climate has been taken. The reconstructions are given in fractional coverage per area [%].

| no. | biome | dominant pft | subpft | additional environmental limits |
|---|---|---|---|---|
| 1 | Tropical evergreen forest | tropical evergreen trees | - | - |
| | | tropical deciduous trees | - | number of green days > 300 |
| 2 | Tropical semi-deciduous forest | tropical deciduous trees | - | number of green days between 250 and 300 |
| 3 | Tropical deciduous forest/woodland | tropical deciduous trees | - | number of green days < 250 |
| 4 | Temperate deciduous forest | temperate deciduous trees | No temp. broadleaved or boreal evergreen tree present | - |
| | | temperate deciduous trees | Boreal evergreen trees present | Twm > 21 |
| | | boreal evergreen trees | temp. deciduous trees present | GDD5 > 900 and Tcm > -19, Twm > 21 |
| | | Boreal deciduous trees | Temp. deciduous trees | |
| 5 | Temperate conifer forest | cool conifer | No temp broadl. trees present, no boreal dec. trees subdominant | - |
| | | cool conifer | temperate deciduous trees with nearly similar npp | - |
| 6 | Warm mixed forest | temperate broadleaved trees | | - |
| | | Temperate deciduous trees | No boreal trees, but temp. broadl-leaved trees present | |
| | | temperate deciduous trees | No boreal trees, but cool conifer present | Tcm > 3 and GDD5 > 3000 |
| | | cool conifer | temp. broadleav. trees present | - |
| 7 | Cool mixed forest | temperate deciduous trees | boreal evergreen trees present | Twm<21 and Tcm > -15 |
| | | boreal evergreen trees | temp. deciduous trees present | GDD5 > 900 and Tcm > -19, Twm < 21 |
| 8 | Cool conifer forest | boreal evergreen trees | no temp. dec. trees present | GDD5 > 900 and Tcm > -19 |
| 9 | Cold mixed forest | temperate deciduous trees | boreal evergreen trees present | Twm<21 and Tcm < -15 |
| | | cool conifer | boreal deciduous trees | - |
| | | boreal evergreen trees | temp. deciduous trees present | GDD5 < 900 and Tcm < -19 |
| | | boreal deciduous trees | cool conifer | - |
| | | boreal deciduous trees | - | GDD5 > 900 and Tcm > -19 |
| 10 | Evergreen taiga/montane forest | boreal evergreen trees | no temp dec trees present | GDD5 < 900 and Tcm < -19 and npp > 350 |
| | | boreal deciduous trees | boreal evergreen trees | - |
| 11 | Deciduous taiga/montane forest | boreal deciduous trees | no temp dec. or cool conifers | GDD5 < 900 and Tcm < -19 |
| 12 | Tropical savanna | shrubs | tropical trees present | woody LAI > 3.5 |
| 13 | Temperate broadleaved savanna | shrubs | temp. deciduous. trees present | |
| 14 | Temperate sclerophyll woodland | shrubs | temp.broadleav. trees present | - |
| 15 | Temperate xerophytic shrubland | woody desert | - | grass LAI>1.2 and Tmin < 0 |
| 16 | Tropical xerophytic shrubland | woody desert | - | grass LAI > 1.2 and Tmin > 0 |
| | | shrubs | tropical trees present | woody LAI<3.5 |
| 17 | Open conifer woodland | shrubs | cool conifer present | - |
| 18 | Boreal parkland | boreal evergreen trees | - | GDD5 < 900 and Tcm < -19 and npp < 350 |
| | | shrubs | boreal trees present | Twm < 21 |
| | | Boreal deciduous trees | - | NPP < 250 |
| 19 | Tropical grassland | C4 tropical grass | - | - |
| 20 | Temperate grassland | C3/C4 temperate grass | - | GDD0 > 800 |
| 21 | Desert | woody desert | - | grass LAI < 1.2 |
| | | temp. or trop. Trees or conifer | - | NPP < 100 |
| | | C3/C4 temperate grass | no boreal trees possible | - |
| 22 | Steppe tundra | C3/C4 temperate grass | - | GDD0 < 800 |
| | | cold herbaceous | - | - |
| 23 | Shrub tundra | tundra shrub | - | GDD0 > 650 |
| 24 | Dwarf shrub tundra | tundra shrub | - | 400 < GDD0 < 650 |
| 25 | Prostrate shrub tundra | tundra shrub | - | GDD0 < 400 |
| 26 | Cushion forb lichen moss tundra | lichen / forb | - | - |
| 27 | Barren | no pft possible | - | - |
| 28 | Land ice | all | all | Twm<2 |

**Figure A1: Biome assignment rules in BIOME4. Listed are the dominant and subdominant plant functional types as well as additional environmental limits of each biome. Modifications to the original BIOME4 code are marked in red.**

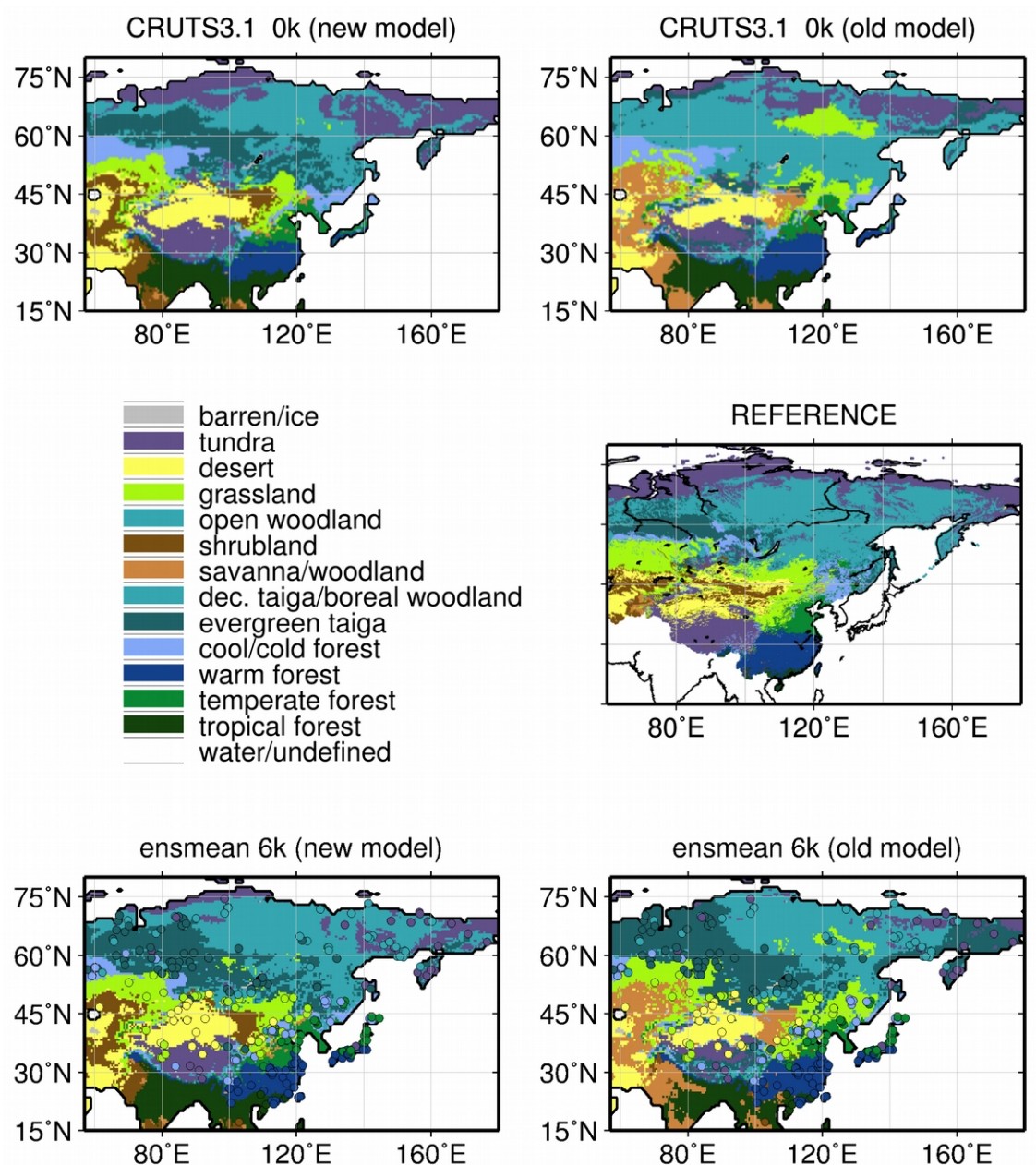

1015 Figure A2: Biome distributions for the modern climate (CRU TS3.10 0k) simulated in the modified BIOME4-version (new model) and the original BIOME4 model (old model), modern biome distribution based on the reference dataset (modified from Hou, 2010; Saandar and Sugita, 2004; and Stone and Schlesinger, 2003) and biome distribution for the mid-Holocene time-slice (6k) based on the ensemble mean climate, using the modified model version and the original model. In the 6k plots, the reconstructed biome distribution (from BIOME6000 project, http://www.bridge.bris.ac.uk/resources/Databases/BIOMES_data; Prentice et al., 2000; 1020 Harrison et al., 2001; Bigelow et al., 2003) is additionally shown for model validation (dots).

100

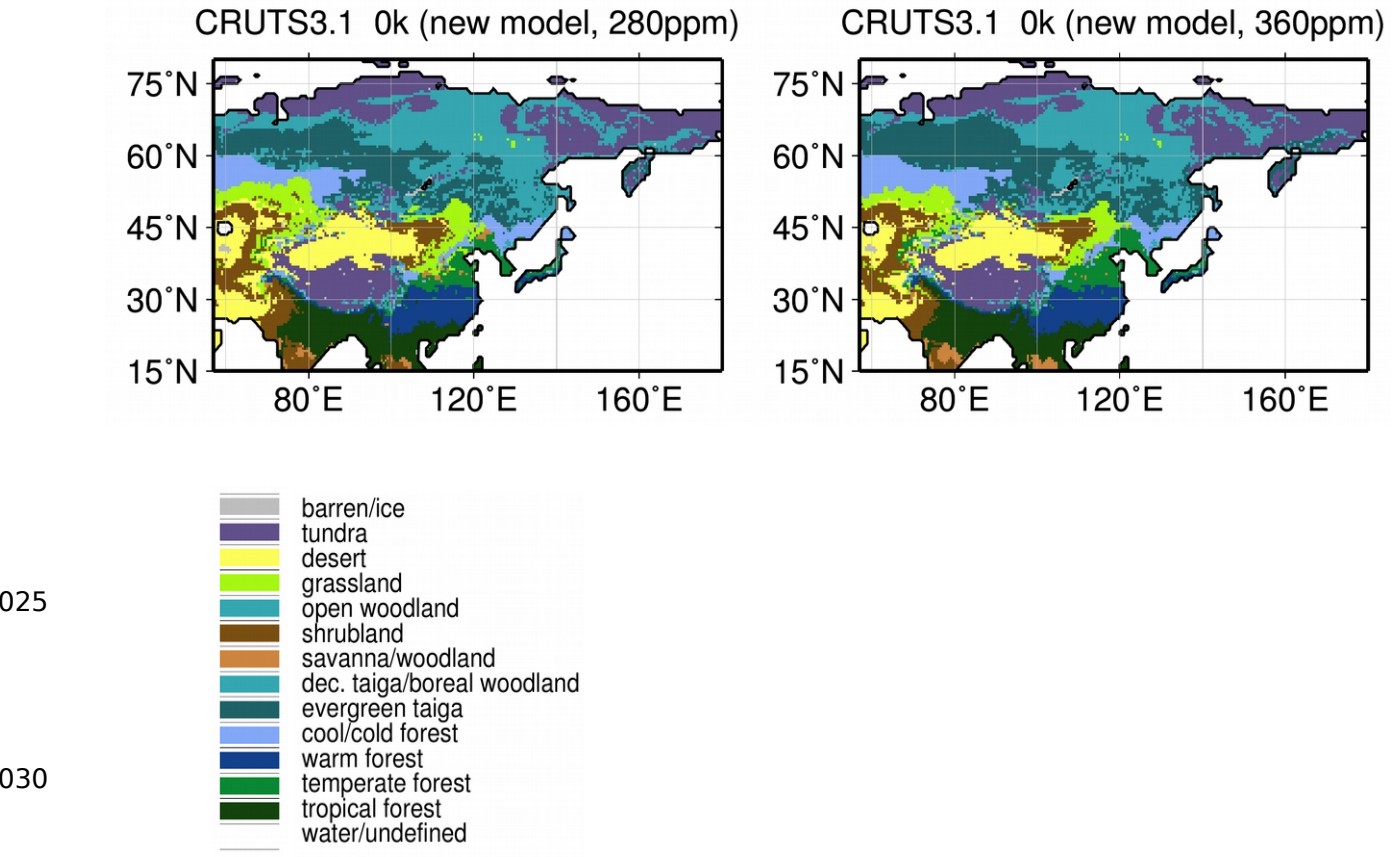

**Figure A3: Biome distributions for the modern climate (CRU TS3.10 0k) simulated in the modified BIOME4-version (new model) using atmospheric $CO_2$-level of 280ppm (pre-industrial) and 360ppm (modern), respectively.**

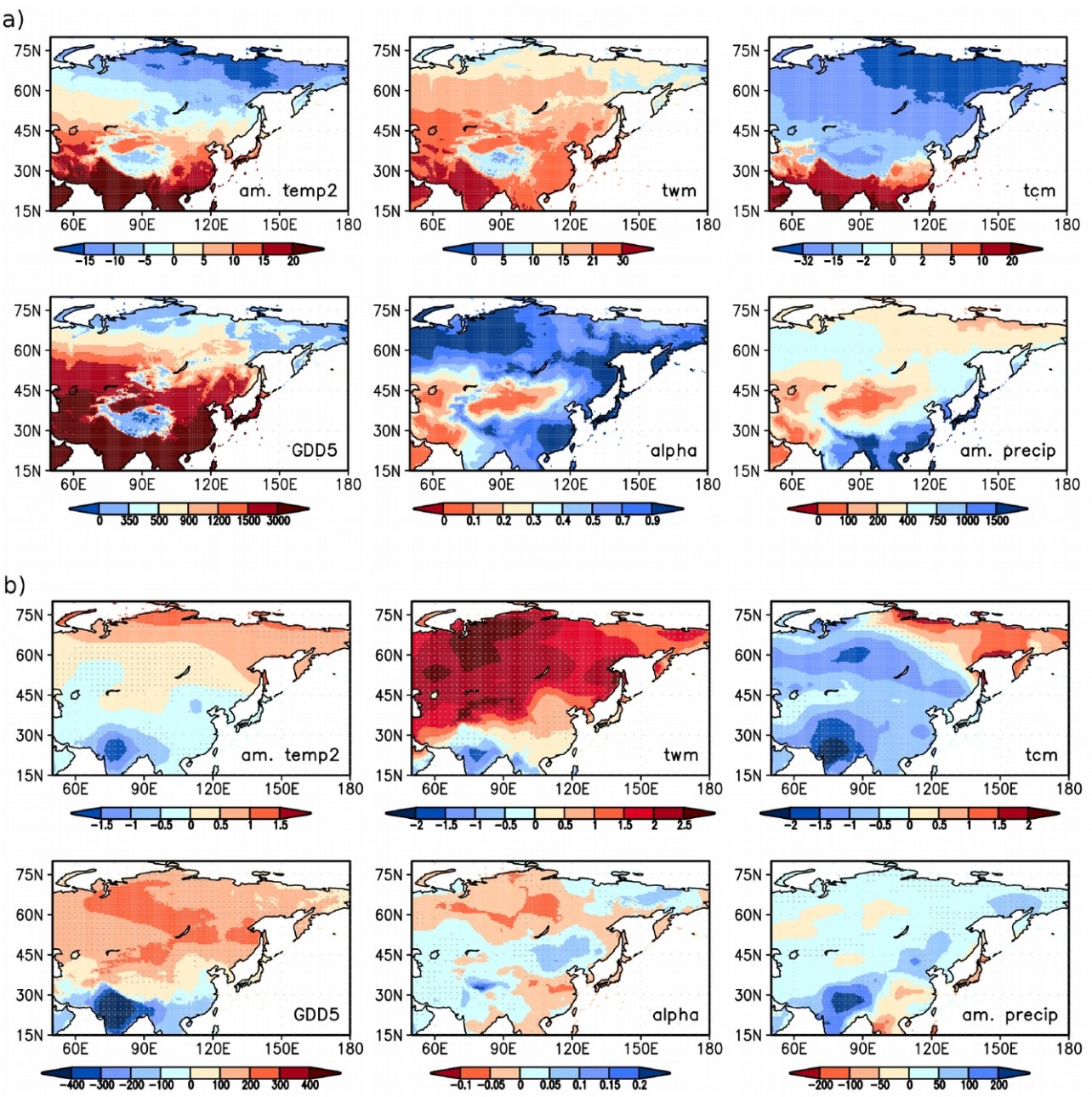

**Figure B1: Simulated bioclimate, i.e. annual mean near-surface air temperature (am. temp2), mean near-surface air temperature of the warmest month (twm) and of the coldest month (tcm), growing degree days on a basis of 5° (GDD5), Taylor-Priestley coefficient of annual actual to equilibrium evapotranspiration (alpha) as calculated in Prentice et al. (1992) and the annual mean precipitation (am. precip), a) based on modern climatology (CRU TS3.10) and b) difference of the ensemble mean bioclimate at mid-Holocene and the modern climatology (6k-0k). Temperature and GDD5 are given in °C, precipitation is given in mm/year.**

1040

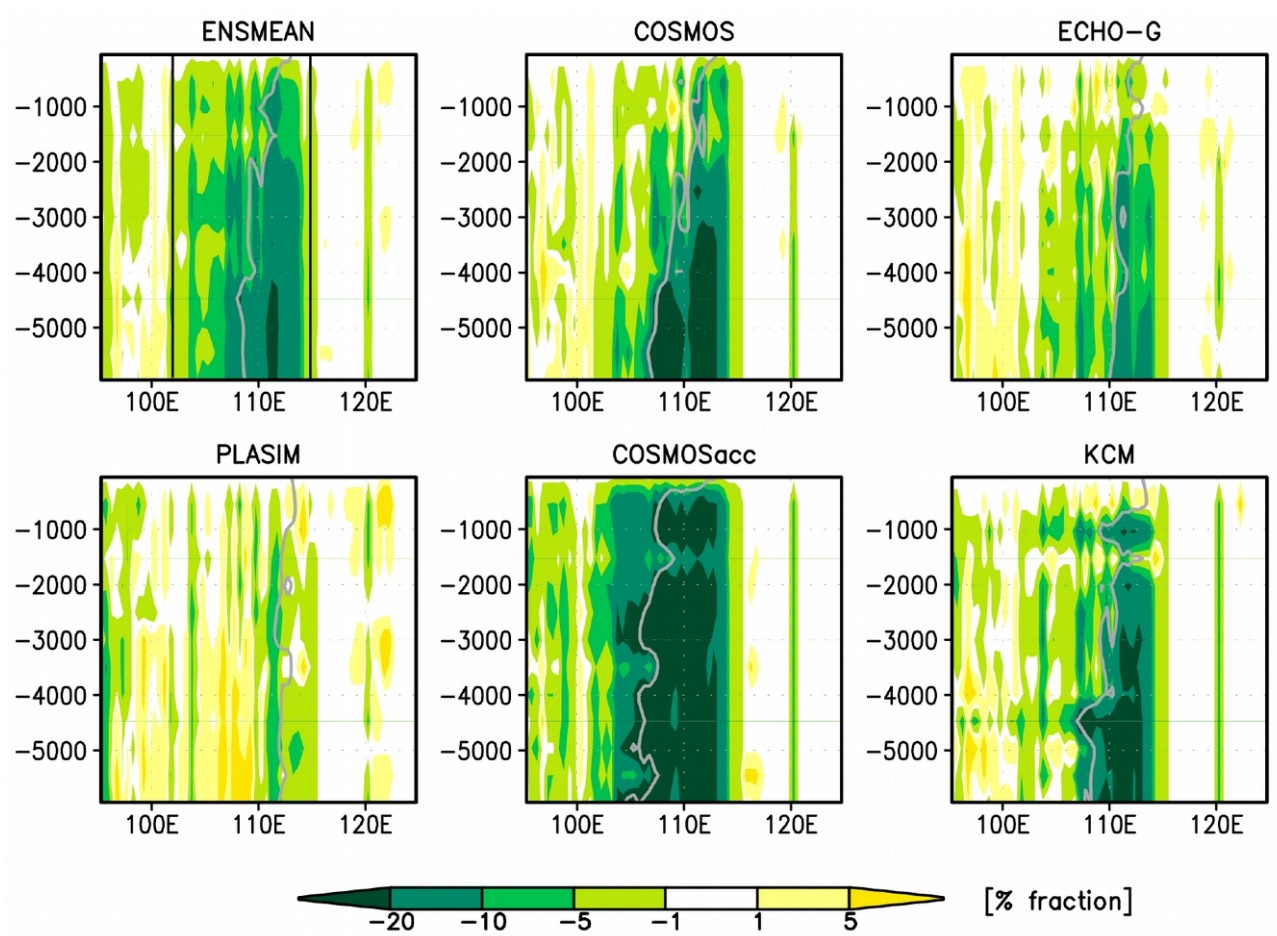

**Figure C1: Mid-Holocene to pre-industrial desert biome change in the desert-steppe-forest transition zone (95°-125°E, 32°-52°N, land only). Hovmoeller diagrams showing the absolute change in meridional mean desert fraction [%] compared to pre-industrial as ensemble mean and for the individual simulations. The grey lines mark the steppe border (here ad hoc defined as the isoline of 20% desert fraction) the black lines mark the regions with strongest precipitation change (c.f. Fig.6).**

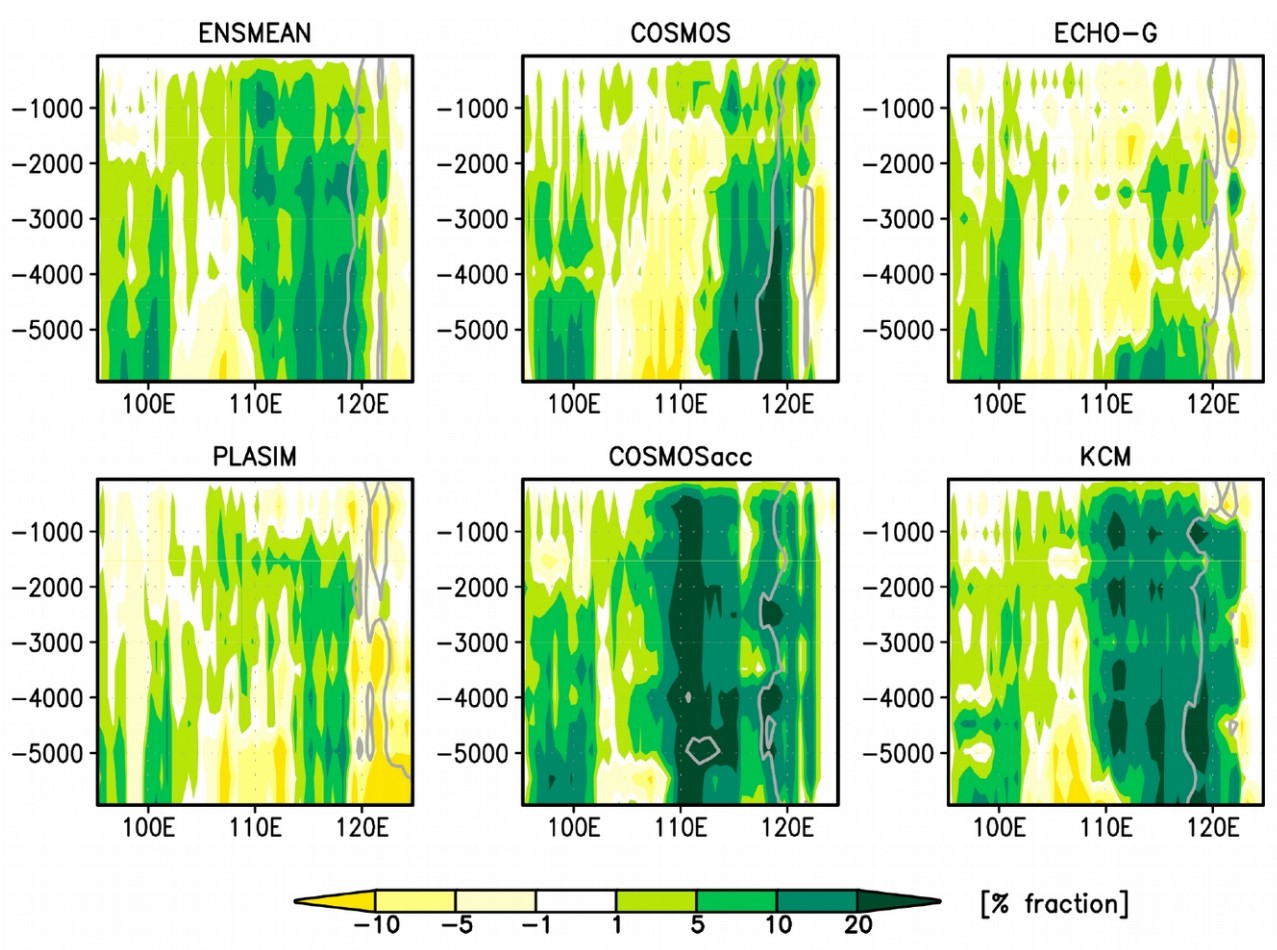

1050 Figure C2: Same as Fig.C1, but for forest fraction in the desert-steppe-forest transition zone The grey lines display the steppe-forest margin (here ad hoc defined as isoline of 80% forest fraction).

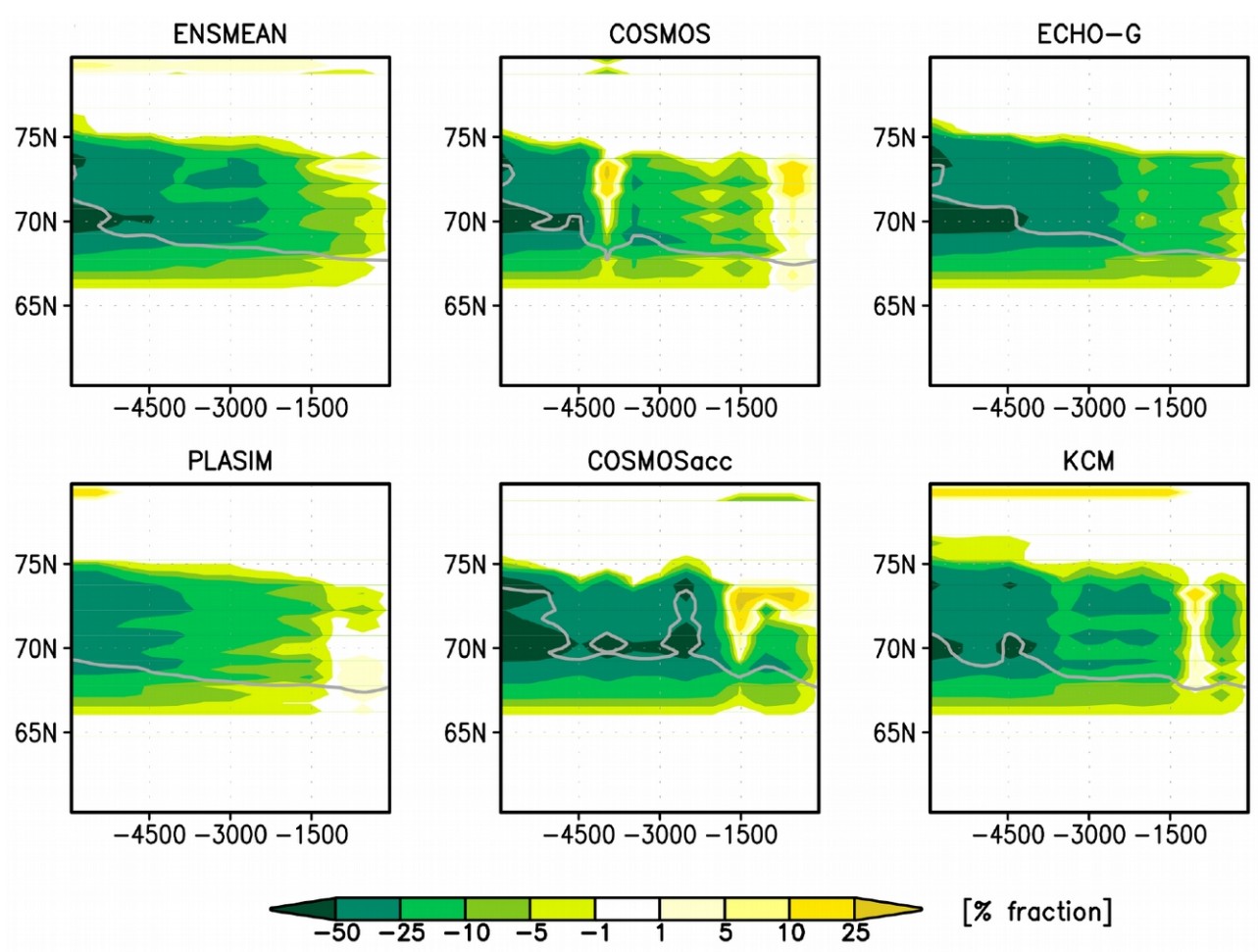

**Figure C3: Same as Fig.C1, but for the tundra fraction in the north-central Siberian taiga-tundra margin (66°-120°E, 60°-80°N).**
**The grey lines display the taiga-tundra margin (here ad hoc defined as isoline of 20% tundra fraction).**