# Peer review of "Biome changes in Asia since the mid-Holocene – an analysis of different transient Earth system model simulations"

_Climate of the Past, 2016_

## Referee Comment (RC1) · Anonymous Referee #1 · 13 Sep 2016

The manuscript of Dallmeyer et al. applied BIOME4 model, driven with different transient Earth system model simulations, to get the general potential vegetation changes from Holocene to 0K in Asia. Model experiments are well organized, including the modern validation, sensitivity test, anomaly approach for climate model bias treatment, general pattern analysis, and comparison for the key region using pollen-reconstructed vegetation transient changes. The paper is well written, with clear thoughts and solid results. Thus, I rate it a good manuscript, and I have the following issues, hope the author could address them.

1 Model tuning/validation are using inconsistent information for 0K reference run, e.g. period for climate data, $CO_2$ concentration, and biome. Author chose the averaged

climatology of 1960-2000 from CRU TS3.1 dataset, and drove BIOME4 model with preindustrial 280 ppm CO2-concentration, rather than the mean CO2 concentration of 1960-2000. However, the reference observed biomes for comparison is from modern data. To be simplify, author is using Modern climate + PI CO2 to simulate Modern biomes. I understand that author hope to show model's performance in lower CO2 environment. However, in BIOME4 model, CO2 has been treated as a very important factors for biome distribution. Thus you cannot ignore the effect of change CO2 concentration from 280ppm to 345ppm or the mean level during 1960-2000.

It would be helpful and more convinced if the author could add the following information:

a) A detailed logic about using preindustrial CO2 concentration (280 ppm) and modern climate, but comparing with modern biome data;

b) Quantify the CO2 effects on modern biome distribution by comparing simulation using mean CO2 concentration during 1960-2000 with simulation using prescribed 280ppm. The similarity of the two simulations would support the choice of using 280 ppm CO2 concentration.

c) If using modern CO2 concentration did generate some difference, would it affect the adjustment of BIOME4 model, e.g. LAI, NPP and soil moisture limits, and other bioclimate limits?

I think this is very important. Because here the model is highly tuned with today's observation. The choice of modern climate and CO2 data would affect this tuning. The change of those modified climate (and other) limits would have impact on the transient MH simulation.

2 Author mentioned that BIOME4 is an equilibrium vegetation model, and also attributed the failure of simulating modern Arctic tundra with deciduous taiga and boreal woodlands by this equilibrium issue. This also caused the inconsistent short-term variability between simulation and pollen-reconstructed biomes. Does this mean the

climate change during each 500 years are either too big variation or too short period to be treated as equilibrium? And how big the influence is for the transition zone simulation?

3 One of the main targets of this paper is "to test the robustness of the simulated vegetation changes and quantify the differences among the models". As pollen-reconstructed biomes can be used another way for climate model-data comparison, it would be much more interesting if some more analysis/discussion about the evaluation/quantification about effects from uncertainties and bias of climate models in terms of model components, forcing setting. And this would also be more relative to this paper's subtitle "an analysis of different transient Earth system model simulations".

Other questions and technical issues:

1 in Figure2, it is clear that there is discrepancy for desert, shrubland, and grassland between observation and simulation for desert-stepper-forest transition zone (Region 1 in Figure1). Would it have effects on the 6k simulation?

2 When the GCM output is interpolated into 0.5, whether elevation is considered? Would elevation have impacts on BIOME4 simulations using different GCM model outputs. Beacause different models have different spatial resolutions, which would deliver different elevation to the same 0.5 grid cells.

3 500-year window length was applied for time slice analysis, and 120-year long-term mean was used to represent the climate status for each time slice. However, it is still not very clear, whether the 120 years are starting from the first year of each window or these 120 years are evenly distributed around the starting year of the window.

4 In Figure6b, how can the precipitation be negative value?

5 And please also check the data for Figure 6. The ensemble mean precipitation of 30mm/day for a desert-stepper-forest transition zone should be too big.

6 line 472, could the author explain the meaning of temporal "linear", and why should

we expect it should be linear or non-linear?

---

## Referee Comment (RC2) · Anonymous Referee #2 · 28 Sep 2016

In this manuscipt, Dallmeyer et al. use BIOME4 to simulate Asian vegetation changes from the mid-Holocene (6 ka BP) to present. They first calibrated the model to better simulate the present vegetation in the Asian region, and then applied it to the Holocene. To assess the uncertainties in simulating Holocene vegetation, the authors use the output from 5 transient Holocene simulations with different earth system models (ESM) as the climate forcings for BIOME4. The BIOME4 results indicate an easterward (southward) shift of the desert-steppe-forest (forest-tundra) boundary in the transitional monsoon zone (Arctic) from the mid-Holocene to present. This trend is robust in spite of the climate forcings used for BIOME4, and is in general consistent with the proxy data retrieved from these regions. However, there are large uncertainties in the magnitude

and temporal variation of these vegetation shifts among the BIOME4 simulations using different ESM output, which is difficult to be compared with the available proxy data. Some discussions on these issues are offered in the manuscript. In general, the manuscript is innovative in its effort of using multi-ESM output for modeling vegetation and assessing model uncertainties, and also in its effort of modeling vegetation changes throughout the mid-Holocene which can be compared with proxy records on time dimension. The manuscript is written in a clear and logic way, but there are still some aspects that may be improved.

Major comments:

1. A clearer explanation of the rational of the methods is needed:

(1) What's the resolution of BIOME4? Although we could speculate the resolute of BIOME4 in this study is 0.5x0.5 degree according to the resolution of CRU TS3.1, it should be better if the author could state this clearly and explain why 0.5x0.5 degree is sufficient? Would higher resolution of BIOME4 be beneficial?

(2) It is said that "The differences between the monthly mean climatologies (long-term averages of 120 years) simulated for each time-slice and the simulated pre-industrial climate have been added to the reference dataset. " How do authors calculate the differencse? Is it absolute difference or relative difference? Do the author use the same methods for all the climate variables?

(3) Some of the ESMs used in the study also include dynamic vegetation (e.g., CO-MOS, COSMOSacc, and PLASIM). Have the authors checked how the vegetation changes in these fully couple runs? Are they consistent with the offline simulations using BIOME4? BIOME4 simulations using output from COSMOS and COSMOSacc seem to exhibits the largest difference between mid-Holocene and present day. Is this partially related to the fact that vegetation feedbacks have been included in COSMOS and COSMOSacc runs?

2. I suggest more discussion on the temporal variation of Asian vegetation in the proxy and in the model, which is one of the key aspects in this study.

(1) The decline of forest biome from 6k to 0k in Daihai record can be related human activity. The author should separate possible human induced changes from climate driven changes so as to better compare with the model results which is purely climate driven changes. For instance, based on Figure 11, the proxy record implies a strong decline of forest biome from 3500 to 2500 BP in Daihai. Can this be caused by human activity?

(2) The potential linkage of vegetation changes between the monsoon region and the Arctic region. From mid-Holocene to present, a southward shift of forest-tundra boundary in the Arctic (colder Arctic) corresponds to a eastward shift of desert-steppe-forest boundary in the transitional monsoon region (weaker East Asian summer monsoon). Does this relation between Arctic and monsoon vegetation change also exist in a shorter time scale in both proxy and the model results?

(3) Why the temporal changes of vegetation from 6k to 0k is non-linear in both proxy and model results? Does this related to orbital forcing or internal feedbacks?

(4) The shift of desert-steppe-forest boundary in the transitional monsoon region is shown to be linked to precipitation changes in this region, but is it also linked to East Asian summer monsoon strength? It would be interesting to see if the shift of desert-steppe-forest boundary is in line with the changes of the East Asian summer monsoon strength from 6k to 0k.

Specific comments:

Introduction: An introduction of existing vegetation simulations for the mid-Holocene in Asia may be useful.

Line 200: "medium" should be "intermediate"

Line 277-278: Why cold season temperature decrease in the tropical region in mid-

Holocene?

Line 327-329: rephrase the sentence to be clearer: Should "westward of 118" be "eastward of 118"?

Line 342, Line770: "Fig.D" does not exist?

Line 418-419: references need to be added for the statement

Summary and Conclusion: can be shortened.

Table 3: Please add information on whether the models use dynamic vegetation.

Figure1: Please explain the rational for the lightblue line in the figure? How do you define the extent of the Asian monsoon region? Please specify which climate dataset or reference the summer circulation at 850 hPa in the figure is based on? The sketch of the summer circulation may be oversimplified and thus misleading to the reader.

Figure2: Why there is no vegetation cover over India and South East Asia in the reference map? Please explain.

Figure6,7,8,9,10: Please use different color or line type to represent each individual simulations.

Figure9: While the caption says (a) is absolute difference in annual mean temperature, the figure title of (a) have unit "%" suggesting relative difference? Which one is true?

Figure11: The label of time axis is better to be consistent with previous figures (using minus values for year?). It would be better if each individual simulation can be shown in the figure to see which simulation is in the best agreement with the proxy data.

---

## Referee Comment (RC3) · Anonymous Referee #3 · 28 Sep 2016

General comments:

Thank you for putting together all the material.

The first part of this paper describes biome changes over the entire Asia region between the mid-Holocene and pre-industrial period using several different transient climate simulations. Through sensitivity experiments, the authors also investigate the causes of the biome changes in Asia. The second part of this paper describes biome changes at two transition zones (between forest-type and non-forest-type biomes) in Asia over the last 6000 years using the transient climate simulations and a few pollen-based biome reconstructions. In order to simulate biomes in Asia, the authors use a slightly modified BIOME4 equilibrium vegetation model. As a result, most of the results

depend on the BIOME4 model.

Although the authors slightly modified and recalibrated the BIOME4 model for pre-industrial CO2 concentration (280 ppm), readers may not understand how to validate the model. Therefore, the authors need to validate the modified BIOME4 model performance quantitatively. I think this is a key process for this study because your study really depends on the vegetation model. Moreover, if possible, the authors should use more than one model in order to reduce the dependence of the results on the choice of vegetation model.

The authors select one pollen-based biome reconstruction for each target transition zone, but the data does not represent sufficiently the feature of the large regions. Therefore, if possible, the authors should use more than one data for the target regions. You might download more pollen data from Neotoma database (http://www.neotomadb.org).

Despite of these remarks, I think that it is a very interesting paper.

Specific comments

L 1. The title might be a little vague for me because your target period is during the last 6000 years, not the entire Holocene.

L 31. "since the mid-Holocene"? for "during the Holocene". Your study focuses on the changes in climate and vegetation over the last 6000 years, not the entire Holocene.

L 41. "during the Holocene", As mentioned before, do you check the entire Holocene?

L 80. How many plant functional types (PFTs) do we need to describe the vegetation in Asia? How many PFTs are used in the current Earth System models? Does BIOME4 vegetation model have enough PFTs for the aim?

L 91. Why do you choose BIOME4 vegetation model? Can you choose other vegetation model(s) in this study?

[Figure]

L 115. "280 ppm" for "280ppm"

L 115-129. How do you calibrate and validate the BIOME4 model for pre-industrial CO2 concentration (280 ppm)? You should show the modified BIOME4 model performance quantitatively using the any data/observations. I do not know whether the model works better or not in the Figure 2, map-map comparison.

L 123. "1200 °C" for "1200°C"

L 125. "280 ppm" for "280ppm"

L 126. "CRU TS3.10" for "CRU TS3.1"; why don't you use a newer reference climate data (e.g., CRU TS3.21 or TS3.22)?

L 132. "0 ka" or "0 ka BP" for "0k" because of the consistent abbreviation between model and data (?)

L 135-139. As mention before, you have to show your modified BIOME4 model performance quantitatively. Compared to the original BIOME4 model, does your BIOME4 model simulate a better biome distribution in Asia? Why don't you use BIOME 6000 data for your model validation?

L 165. "6000 years ago, 6 ka" for "6000 years before present (henceforth referred to as 6k)" or "6 ka BP" for "6k" because of the consistent abbreviation between model and data (?)

L 226. "The simulated biomes"? for "the model-based biome reconstructions"

L 241. You should evaluate the mid-Holocene biome distribution using BIOME 6000 data in Asia because we do not know your model results are consistent with observed data or not.

L 243-261. How do you choose the target regions (95-125°E, 32-52°N) and (60-180°E, 15-80°N)? I mean Figure 3 also show large vegetation changes occur at eastern Siberia (tundra vs. taiga) and west-central Asia (60-80°E, 50-60°N; grassland vs.

cool/cold forest).

L 254. If possible I would like to see the Figure 4 information using the 0 ka simulated biomes and reference data. Which is a larger differences of biomes between 6 ka/ vs. 0 ka and 0 ka vs. reference data?

L 271-272. The results from the sensitivity experiments show the real vegetation response or just BIOME4 response? If we use different vegetation models, do we get different results, for example cloud cover is a key factor of vegetation changes (it is opposite to L 275)? Nemani et al. (2003, Science) also shows the similar results about geographic distribution of potential climatic constraints to plant growth.

L 357. About Figure 11, less/more what? What does the x-axis shows, fractional changes in biome or ...?

L 357. Why do you use only two reconstructions? Can you use more pollen data from Neotoma database (http://www.neotomadb.org)?

L371. "The overall change in biome composition since the mid-Holocene"? for "The overall change in the Holocene biome composition"

L 380-383. You should consider the vegetation model deficits too. Your results really depend on BIOME4 vegetation model.

L 398. "Local"? for "regional"

L 405-412. Even if vegetation reconstructions for Asia are sparse, you should use available data (i.e. BIOME 6000 data) for your simulated biomes at the pre-industrial and mid-Holocene first.

L 410-412. Please put any references.

L 431. Your simulations focus on the last 6000 years, not "during the Holocene"

L 439. "500 years" for "500years"

L 496. "More pollen records are needed to evaluate the simulated results." I understand your argument, but please use available dataset and show us the results first.

---

## Author Comment (AC1) · 11 Nov 2016

Referee #1 (R1)

R1: "Model tuning/validation are using inconsistent information for 0K reference run, e.g. period for climate data, CO2 concentration, and biome. Author chose the averaged climatology of 1960-2000 from CRU TS3.1 dataset, and drove BIOME4 model with pre-industrial 280 ppm CO2-concentration, rather than the mean CO2 concentration of 1960-2000. However, the reference observed biomes for comparison is from modern data. To be simplify, author is using Modern climate + PI CO2 to simulate Modern biomes. I understand that author hope to show model's performance in lower CO2 environment. However, in BIOME4 model, CO2 has been treated as a very important factors for biome distribution. Thus you cannot ignore the effect of change CO2 concentration from 280ppm to 345ppm or the mean level during 1960-2000. It would be helpful and more convinced if the author could add the following information:

a) A detailed logic about using pre-industrial CO2 concentration (280 ppm) and modern climate, but comparing with modern biome data

A: We fully agree, that this is misleading. There are several reasons why we chose this constellation. a) modern climate data is the only observed (and model independent) data that exist and we want to have a reliable reference climate. b) Most climate models fixed the CO2 concentration to pre-industrial values (i.e., 280ppm) and we want to calculate the vegetation distribution consistent to the climate data c) we compared the reference simulation with potential vegetation, that is probably not in equilibrium with the current, fast changing, atmospheric CO2-level d) Simulations using CO2-level of 280ppm or 360ppm differ only slightly on Macro-Biome-level (at 2108 out of 33800 grid-boxes, i.e.6.24% of the grid-boxes).

We have revised our manuscript by writing in the chapter about the reference simulation: "As reference simulation for the modern biome distribution (named pre-industrial or 0k in the following), we forced BIOME4 with the modern monthly mean climatology (1960-2000) taken from the University of East Anglia Climatic Research Unit Time Series 3.10 (CRU TS3.10, University of East Anglia, 2008, Harris et al., 2012) providing a more reliable climate background than pre-industrial climate reconstructions or simulations. Though, we prescribed pre-industrial atmospheric CO2-concentration (280 ppm) to be consistent to the transient Holocene climate simulations and to partly come up with the fact, that modern vegetation is supposed to be not in equilibrium with the fast changing atmospheric CO2-level. The differences between the reference simulation using 280 ppm and a simulation prescribing 360 ppm can be seen in the Appendix (Fig. A3)."

R1: "b) Quantify the CO2 effects on modern biome distribution by comparing simulation using mean CO2 concentration during 1960-2000 with simulation using prescribed 280ppm. The similarity of the two simulations would support the choice of using 280 ppm CO2 concentration. "

A:On Macro-Biome-level, simulations with 360ppm or 280ppm differ only slightly. In the region of interest, the simulation with 360ppm shows slightly more temperate forest in Eastern China and also more cool forest in Central-Western Asia. This has no effect on the results and conclusions of our manuscript. We added a figure showing the simulated 0k biome distributions using 280ppm and 360ppm, respectively, to the Appendix (Fig. A3)

R1: "c) If using modern CO2 concentration did generate some difference, would it affect the adjustment of BIOME4 model, e.g. LAI, NPP and soil moisture limits, and other bioclimate limits? I think this is very important. Because here the model is highly tuned with today's observation. The choice of modern climate and CO2 data would affect this tuning. The change of those modified climate (and other) limits would have impact on the transient MH simulation."

A: The choice of using 280ppm has no effect on the general conclusions drawn in this manuscript. Most of the slight adjustments (such as bioclimatic limits) are independent of the CO2-level. Therefore, we do not expect an impact on the results for the mid-Holocene vegetation change.

R1: "Author mentioned that BIOME4 is an equilibrium vegetation model, and also attributed the failure of simulating modern Arctic tundra with deciduous taiga and boreal woodlands by this equilibrium issue. This also caused the inconsistent short-term variability between simulation and pollen-reconstructed biomes. Does this mean the climate change during each 500 years are either too big variation or too short period to be treated as equilibrium? And how big the influence is for the transition zone simulation¿'

A: We are not sure, if we understand this comment correctly. The short-term variability revealed by the reconstructions can not be seen in the model, because the temporal

resolution of the BIOME4 simulations are 500 year time-slices, only. The here used reconstructions have a much finer resolution. The climate forcing data for the time-slices are considered to be at equilibrium conditions.

R1: "One of the main targets of this paper is "to test the robustness of the simu-lated vegetation changes and quantify the differences among the models". As pollen-reconstructed biomes can be used another way for climate model-data comparison,it would be much more interesting if some more analysis/discussion about the evalua-tion/quantification about effects from uncertainties and bias of climate models in terms of model components, forcing setting. And this would also be more relative to this paper's subtitle "an analysis of different transient Earth system model simulations".

A:We agree, that more investigations on the uncertainties and biases related to the dif-ferent model forcing and model components would be interesting, but to quantify this, more simulations (using different model components and forcing) are needed, which are very expensive and consume much computational power. We added a short dis-cussion on the differences in interactive model components between the models and their potential effects on vegetation simulation: 'At least partly, the discrepancies in simulated climate may be related to the differences in interactive model components used in the climate model, i.e. some models include dynamic vegetation some do not. To test the influence of dynamic vegetation on the simulated Asian climate, sensitiv-ity simulations have to be undertaken. An appropriate set of experiments only exists in the COSMOS-setup (cf. Dallmeyer et al., 2010.). According to these simulations, interactive vegetation has a negligible effect on the mid-Holocene to pre-industrial pre-cipitation change in the desert-steppe-forest transition zone. Vegetation feedbacks contribute to the warmer mid-Holocene climate in the high northern latitudes, but the interactive ocean has a much stronger impact on the climate change. Therefore, the lack of interactive vegetation in KCM and ECHO-G may partly lead to biases in simu-lated climate change, but we do not expect an effect of this on the general results of this study.'

R1: "in Figure2, it is clear that there is discrepancy for desert, shrubland, and grassland between observation and simulation for desert-stepper-forest transition zone (Region 1 in Figure1). Would it have effects on the 6k simulation¿'

A: The main difference in the 0k simulation is the occurrence of a dry shrubland belt between the desert region and the steppe-region. In the observations, this shrubland belt is rather located in between the desert region. This 'separated' biome belts are also seen in the 6k simulations. However, as we group the biomes 'dry shrubland' and 'desert' into the Macro-Biome 'desert', and discuss the transient biome changes on the basis of the Macro-Biomes, only, we do not expect an effect of the differences between the reference dataset and the 0k model simulation on the the general results and conclusions of this study.

R1: "When the GCM output is interpolated into 0.5, whether elevation is considered? Would elevation have impacts on BIOME4 simulations using different GCM model outputs. Because different models have different spatial resolutions, which would deliver different elevation to the same 0.5 grid cells."

A: As we use the anomaly method to calculate the biome distribution, the elevation - or more precisely – the climate gradients resulting from the elevation are preserved from the CRU reference climate. Implicit in this widely used downscaling approach is the assumption that the temperature lapse rate does not change with climate. This is a fair assumption, if not drastically different climate states (e.g. glacial vs. interglacial climate) are considered. However, different spatial resolutions lead to different representations of the elevation in the models, which may have an impact on the climate change during the Holocene. We added this information to the Discussion (ll 422-427).

R1: "500-year window length was applied for time slice analysis, and 120-year long-term mean was used to represent the climate status for each time slice. However, it is still not very clear, whether the 120 years are starting from the first year of each window or these 120 years are evenly distributed around the starting year of the window.

A: We agree, this is not mentioned in the text. We use the years 1 – 120 (or 12) of each climate model simulations for 6k, year 501-621 (50-61) for 5.5k, etc. and the last 120 (12) years for 0k. We added this information in the revised manuscript (ll.226-227).

R1: "In Figure6b, how can the precipitation be negative value¿'

A: "The figure shows the differences in precipitation to the pre-industrial value"

R1: "And please also check the data for Figure 6. The ensemble mean precipitation of 30mm/day for a desert-stepper-forest transition zone should be too big."

A: "Many thanks, the unit was wrong, it is mm/year! We corrected it.

R1: "line 472, could the author explain the meaning of temporal "linear", and why should we expect it should be linear or non-linear¿'

A: We agree, this is misleading, we now write: "The expansion of desert during the Holocene back to modern distribution is not uniform and varies spatially."
* * *
[Figure]

**Fig. 1.** Biome distributions for the modern climate (CRUTS3.10 0k) simulated in the modified BIOME4-version (new model) using atmospheric CO2-level of 280ppm (pre-industrial) and 360ppm (modern), respectively.

---

## Author Comment (AC2) · 11 Nov 2016

Referee #2 (R2)

Major comments:

R2: "A clearer explanation of the rational of the methods is needed: What's the resolution of BIOME4? Although we could speculate the resolute of BIOME4 in this study is 0.5x0.5 degree according to the resolution of CRU TS3.1, it should be better if the author could state this clearly and explain why 0.5x0.5 degree is sufficient? Would higher resolution of BIOME4 be beneficial? "

A: We agree, this is poorly explained in the manuscript. The resolution of BIOME4

depends on the resolution of the input data. The resolution of CRU TS3.10 is 0.5x0.5°. This is the best and highest resolved dataset that is available. We now write in Ch.2.2: "Furthermore, the CRU TS3.10 data is provided in relatively high spatial resolution of 0.5°x0.5°, resolving the climate gradients along the complex Asian orography better than the Global climate model simulations."

R2: "It is said that "The differences between the monthly mean climatologies (long-term averages of 120 years) simulated for each time-slice and the simulated pre-industrial climate have been added to the reference dataset. " How do authors calculate the differencse? Is it absolute difference or relative difference? Do the author use the same methods for all the climate variables ? "

A: It is the absolute difference between the climate of each time-slice and the pre-industrial climate (e.g.6k-0k). We use the same method for all simulations and for all climate variabilities. We added 'absolute' to this cited sentence.

R2: "Some of the ESMs used in the study also include dynamic vegetation (e.g., COSMOS, COSMOSacc, and PLASIM). Have the authors checked how the vegetation changes in these fully couple runs? Are they consistent with the offline simulations using BIOME4? "

A: We tested the consistency for the COSMOS simulation and it appears that in principle, the tendencies of less desert and more grassland and forested area in the desert-steppe-forest transition zone during 6k and more forest in the northern latitudes are the same. However, the vegetation in COSMOS (and also in the other vegetation models) is described in form of few plant functional type fractions whose distribution is not directly comparable with the different biomes. Changes in vegetation are given in changes in cover fraction, only, and not from one vegetation type to another. Construction of a matrix for direct comparison between different vegetation types and biomes used in different models is subject of an ongoing project which, however, is not yet finished.

R2: "BIOME4 simulations using output from COSMOS and COSMOSacc seem to exhibits the largest difference between mid-Holocene and present day. Is this partially related to the fact that vegetation feedbacks have been included in COSMOS and COSMOSacc runs? "

A: This is a very interesting question, but to assess the role of vegetation feedbacks in the mid-Holocene to pre-industrial climate change, sensitivity experiments have to be undertaken. For the COSMOS model, we have analysed this on the basis of simulations with similar setup as in this study (the 6k simulation is the same) for the Asian monsoon region (cf. Dallmeyer et al., 2010: Contribution of oceanic and vegetation feedbacks to Holocene climate change in monsoonal Asia, Clim. Past, 6, 195–218, doi:10.5194/cp-6-195-2010.) Interactive vegetation in the COSMOS model has a negligible effect on the precipitation change in the desert-steppe-forest-transition zone, but contributes to the warmer mid-Holocene climate in the high northern latitudes, but for the high northern latitudes interactive ocean has a much stronger impact. We add this information in the Discussion (ll. 428-435; see also comment to Referee #1).

R2: "I suggest more discussion on the temporal variation of Asian vegetation in the proxy and in the model, which is one of the key aspects in this study. The decline of forest biome from 6k to 0k in Daihai record can be related human activity. The author should separate possible human induced changes from climate driven changes so as to better compare with the model results which is purely climate driven changes. For instance, based on Figure 11, the proxy record implies a strong decline of forest biome from 3500 to 2500 BP in Daihai. Can this be caused by human activity? "

A: Human interference cannot be excluded at this site, but archaeological investigation rather show cultural remains from mid-Holocene warm and humid period, disappearing from Daihai Lake region when the climate has become worse (4300 cal years ago, see Xiao et al., 2004 and references therein). Thus, the forest decline at Lake Daihai may probably results from climate change. Though human impact may have affected local vegetation change since the mid-Holocene in Eastern and Central Asia, overall climate

is assumed to be the major driver of forest decline (e.g. Wang et al., 2010, Cao et al. 2015, Tian et al. 2016) We added to the manuscript: "Human interference cannot be excluded at this site (cf. Xiao et al., 2004), but may have affected vegetation change since the mid-Holocene rather locally. In Eastern and Central Asia, overall climate is assumed to be the major driver of forest decline (e.g. Wang et al., 2010, Cao et al., 2015, Tian et al. 2016)"

R2: "The potential linkage of vegetation changes between the monsoon region and the Arctic region. From mid-Holocene to present, a southward shift of forest-tundra boundary in the Arctic (colder Arctic) corresponds to a eastward shift of desert-steppe-forest boundary in the transitional monsoon region (weaker East Asian summer monsoon). Does this relation between Arctic and monsoon vegetation change also exist in a shorter time scale in both proxy and the model results? "

A: The temporal resolution of the model simulation is not high enough to appropriately analyse this link. Further simulations have to be done to investigate this, but this is not part of this study.

R2: "Why the temporal changes of vegetation from 6k to 0k is non-linear in both proxy and model results? Does this related to orbital forcing or internal feedbacks? "

A: To disentangle the climate response to the orbital forcing and to internal feedbacks further sensitivity simulations have to be undertaken, which is beyond the scope of our study.

R2: "The shift of desert-steppe-forest boundary in the transitional monsoon region is shown to be linked to precipitation changes in this region, but is it also linked to East Asian summer monsoon strength? It would be interesting to see if the shift of desert-steppe-forest boundary is in line with the changes of the East Asian summer monsoon strength from 6k to 0k."

A: We studied the evolution of different monsoon sub-system in another paper using

the same set of climate model simulations (cf.:Dallmeyer et al., 2015).

Specific comments:

R2: "Introduction: An introduction of existing vegetation simulations for the mid-Holocene in Asia may be useful."

A: We agree, an overview of other vegetation simulations for mid-Holocene is interesting, but it is not necessary for the understanding of our manuscript. Therefore, we decided to leave it out and keep the Introduction 'short'.

R2: "Line 200: "medium" should be "intermediate" "

A: In principle, the reviewer is right. PLASIM is considered an Earth system model of intermediate complexity (EMIC), as the authors put their model in the Table of EMICs. We kept "medium", however, as this is the term the authors of PLASIM used in the paper cited.

R2: "Line 277-278: Why cold season temperature decrease in the tropical region in mid-Holocene? "

A: This is probably related to the orbital forcing revealing less winter insolation during mid-Holocene compared to present day.

R2: "Line 327-329: rephrase the sentence to be clearer: Should "westward of 118" be "eastward of 118"? "

A: Yes, it should be eastward. We changed it.

R2: "Line 342, Line770: "Fig.D" does not exist? "

A: Thank you, it is Fig. B3.

R2: "Line 418-419: references need to be added for the statement"

A: We fully agree, and added references to both statements (ll 452-453).

R2: "Summary and Conclusion: can be shortened."

A: We split up the Summary and Conclusion into 2 Chapter.

R2: "Table 3: Please add information on whether the models use dynamic vegetation"

A: done.

R2: "Figure1: Please explain the rational for the lightblue line in the figure? How do youdefine the extent of the Asian monsoon region? Please specify which climate dataset or reference the summer circulation at 850 hPa in the figure is based on? The sketch of the summer circulation may be oversimplified and thus misleading to the reader.

A: "We use the summer (JJA) mean precipitation isohyet of 2mm/day as definition of the Asian (summer) monsoon region. The sketch is based on observations and reanalysis data (GPCP: for precipitation, ERA40 for wind vectors)." We added this information to the caption.

R2: "Figure2: Why there is no vegetation cover over India and South East Asia in the reference map? Please explain."

A: We translated the dataset that where available for us, unfortunately, a dataset for India and South East Asia was not available.

R2: "Figure6,7,8,9,10: Please use different color or line type to represent each individual simulations."

A: It was not the aim of our study to find the "best" model with respect to the Holocene climate or biome change in Asia. We wanted to present a range of possible changes. Therefore we decided not to represent the individual models separately in the climate discussion.

R2: "Figure9: While the caption says (a) is absolute difference in annual mean temperature, the figure title of (a) have unit "%" suggesting relative difference? Which one is

true? "

A: Thank you very much for this hint, it is indeed the absolute difference in temperature, we corrected it in the Figure.

R2: "Figure11: The label of time axis is better to be consistent with previous figures (using minus values for year?). It would be better if each individual simulation can be shown in the figure to see which simulation is in the best agreement with the proxy data."

A: We changed the label of the time axis. It was not the aim of our study to find the "best" model. Therefore we decided not to present the individual models separately.

References: Cao, X., Herzschuh, U., Ni, J., Böhmer, T.: Spatial and temporal distributions of major tree taxa in eastern continental Asia during the Late Glacial and Holocene (22–0 cal ka BP). The Holocene 25, 79-91,2015.

Dallmeyer, A., Claussen, M., Fischer, N., Haberkorn, K., Wagner, S., Pfeiffer, M., Jin, L., Khon, V., Wang, Y., and Herzschuh, U.: The evolution of sub-monsoon systems in the Afro-Asian monsoon region during the Holocene– comparison of different transient climate model simulations, Clim. Past, 11, 305-326, doi:10.5194/cp-11-305-2015, 2015.

Tian, F., Cao, X., Dallmeyer, A., Ni, J., Zhao, Y., Wang, Y., Herzschuh, U.: Quantitative woody cover reconstructions from eastern continental Asia of the last 22 kyr reveal strong regional peculiarities. Quaternary Science Reviews 137, 33-44, 2016.

Wang, Y., Liu, X., Herzschuh, U.: Asynchronous evolution of the Indian and East Asian Summer Monsoon indicated by Holocene moisture pattern in monsoonal central Asia. Earth-Science Reviews 103, 135-153., 2010.

Xiao, J.L., Xu, Q.H., Nakamura, T., Yang, X.L., Liang, W.D. and Inouchi, Y.: Holocene vegetation variation in the Daihai Lake region of north-central China: a direct indication of the Asian monsoon climatic history. Quaternary Science Reviews 23: 1669-1679,

2004.
* * *

---

## Author Comment (AC3) · 11 Nov 2016

Referee #3 (R3)

R3: "In order to simulate biomes in Asia, the authors use a slightly modified BIOME4 equilibrium vegetation model. As a result, most of the results depend on the BIOME4 model.

A: We fully agree, the results depend on the BIOME4 model, but many processes that are included in BIOME4 are implemented in a similar, albeit more complex way, in other vegetation models (e.g. the bioclimatic limits). The variance in the results is received by using (very) different climate input data.

R3: "Although the authors slightly modified and recalibrated the BIOME4 model for pre-industrial $CO_2$ concentration (280 ppm), readers may not understand how to validate the model. Therefore, the authors need to validate the modified BIOME4 model performance quantitatively. I think this is a key process for this study because your study really depends on the vegetation model. "

A: We fully agree, and add a figure in the Appendix showing the 0k reference simulation performed in the modified BIOME4 model and the original BIOME4 model (Fig. A2). Furthermore, we add to the end of Ch2.1 (BIOME4 model description): "The difference between the modified and original BIOME4 model can be seen in the Appendix (Fig. A2) based on the 0k reference simulation and the ensemble mean 6k simulation (including comparison with reconstructions)."

R3: "Moreover, if possible, the authors should use more than one model in order to reduce the dependence of the results on the choice of vegetation model."

A: We agree, that our results are model depending, but unfortunately the variety of vegetation models that are suitable for calculating the mid-Holocene to pre-industrial biome change is small. Other biome models are less complex, and results of vegetations models calculating PFT fractions only would have had to be translated into Biomes, a process that could be an additional source of errors. Therefore, we decided to use BIOME4. Furthermore, BIOME4 has the advantage of needing only few climate variables as forcing.

R3: "The authors select one pollen-based biome reconstruction for each target transition zone, but the data does not represent sufficiently the feature of the large regions. Therefore, if possible, the authors should use more than one data for the target regions. You might download more pollen data from Neotoma database (http://www.neotomadb.org)."

A: High quality data is quite few. For the desert-steppe-transition zone no record is available in the Neotoma database at all. Therefore, we decided to select from all

(for us) available records in the target-regions the record, that shows the best quality with respect to dating and data and that is additionally representative for the regional vegetation changes, as inferred from comparison to literature results. The general vegetation trend indicated by the Dahai record is in line with other records in North central China (cf. Zhao et al, 2009) and the 13-CH-12 record agrees with the other records from the northern Siberian tree line area. We also mention this in the main text, and add the information to the methods section (2.5): "For comparison, representative, high quality (with respect to dating and the data) pollen records covering the last 6000 years have been chosen. For the taiga-tundra-transition zone, a record from a small lake located on the southern Taymyr Peninsula (technical name: 13-CH-12; Klemm et al., 2016) is used, that is in line with the vegetation trend seen at other records located at the Siberian treeline (Pisaric et al., 2000, McDonald et al., 2000, Bigelow et al., 2003). The biome change in the forest-steppe-transition zone is reflected by the record from Daihai Lake in Inner Mongolia (China, 40.5°N; 112.5°E, 1225 m a.s.l.; Xu et al., 2010) that is in line with other records in north central China (Zhao et al., 2009 and references therein)."

Specific comments

R3: "L 1. The title might be a little vague for me because your target period is during the last 6000 years, not the entire Holocene."

A: We agree, and changed the title to: Biome changes in Asia since the mid-Holocene – an analysis of different transient Earth system model simulations

R3: "L 31. "since the mid-Holocene"? for "during the Holocene". Your study focuses on the changes in climate and vegetation over the last 6000 years, not the entire Holocene."

A: We changed this.

R3: "L 41. "during the Holocene", As mentioned before, do you check the entire

Holocene? "

A: done. No, unfortunately most of the simulation do not capture the entire Holocene. Therefore, we decided to start at 6k.

R3: "L 80. How many plant functional types (PFTs) do we need to describe the vegetation in Asia? How many PFTs are used in the current Earth System models¿' Does BIOME4 vegetation model have enough PFTs for the aim? "

A: This is indeed a good question. The current Earth System models uses 2-10 natural PFTs (in the interactive mode), it depends on the model. BIOME4 uses 13 different PFTs. The description of tree-PFTs is similar to other vegetation models, but the advantage of BIOME4 is, that it includes several PFTs representing arctic vegetation (e.g. tundra shrubs, cold herbaceous, lichen/forb) that are only partly considered in the dynamic vegetation models.

R3: "L 91. Why do you choose BIOME4 vegetation model? Can you choose other vegetation model(s) in this study?

A: "Since reconstructions are often presented on taxa or biome level, a biome model is most appropriate for the aim of our study. Other biome models are less complex, so we decided to use BIOME4.

R3: "L 115. "280 ppm" for "280ppm" "

A: done.

R3: "L 115-129. How do you calibrate and validate the BIOME4 model for pre-industrial $CO_2$ concentration (280 ppm)? You should show the modified BIOME4 model performance quantitatively using the any data/observations. I do not know whether the model works better or not in the Figure 2, map-map comparison.

A: We agree, and add a figure in the Appendix showing the 0k reference simulation performed in the modified BIOME4 model and the original BIOME4 model (see comment

above).

R3: "L 123. "1200 C" for "1200C" "

A: done.

R3: "L 125. "280 ppm" for "280ppm" "

A: done.

R3: "L 126. "CRU TS3.10" for "CRU TS3.1"; why don't you use a newer reference climate data (e.g., CRU TS3.21 or TS3.22)¿'

A: done. Since we only use the years 1960-2000, it does not matter, which CRU dataset we choose. The data for these years is supposed to be the same in all CRU-datasets.

R3: "L 132. "0 ka" or "0 ka BP" for "0k" because of the consistent abbreviation between model and data(?) "

A: '0k' in the model and '0 cal ka BP' is not the same, so we keep the abbreviation, but added a notice on this in the method section: Please notice, that the pollen reconstructions are dated in calibrated years before present, i.e. before the year 1950 AD (cal. ka BP), thus the time-step 0 cal. ka BP is not identical with the time-slice '0k' used in the modelling result (i.e. a mean of 120 years).

R3: "L 135-139. As mention before, you have to show your modified BIOME4 model performance quantitatively. Compared to the original BIOME4 model, does your BIOME4 model simulate a better biome distribution in Asia? Why don't you use BIOME 6000 data for your model validation? "

A: We agree, and add a figure in the Appendix showing the 0k reference simulation performed in the modified BIOME4 model and the original BIOME4 model (see comment above). We are currently preparing a new synthesis of Palaeovegetation change in Asia, which will also be biomised (for China cf. Ni et al., 2014: Biome distribution over the last 22,000 yr in China, doi:10.1016/j.palaeo.2014.04.023). We will compare

updated versions of these proxy datasets with vegetation model results in a follow-up manuscript. Since we agree, that a spatial comparison of our model results and the BIOME6000 data will improve the reliability of our results, we add a figure, showing the ensemble mean 6k simulation in the original and the modified model in the Appendix (Figure A2).

R3: "L 165. "6000 years ago, 6 ka" for "6000 years before present (henceforth referred to as 6k)" or "6 ka BP" for "6k" because of the consistent abbreviation between model and data (?)"

A: We kept the term '6000 years before present', because this is commonly used in climate modelling studies.

R3: "L 226. "The simulated biomes"? for "the model-based biome reconstructions"

A: done.

R3: "L 241. You should evaluate the mid-Holocene biome distribution using BIOME 6000 data in Asia because we do not know your model results are consistent with observed data or not."

A: Please, see above.

R3: "L 243-261. How do you choose the target regions (95-125E, 32-52N) and (60-180E, 15-80N)? I mean Figure 3 also show large vegetation changes occur at eastern Siberia (tundra vs. taiga) and west-central Asia (60-80E, 50-60N; grassland vs. cool/cold forest)."

A: We agree, this is not explained in the text. We now write in Ch.2.5: "The simulated biomes were evaluated for key biome transition areas showing the strongest biome change in the model ensemble, i.e. the taiga-tundra transition zone in the high northern latitudes and the forest-steppe-desert-transition zone in north-central China. Since the model fails to appropriately represent modern biome distribution in eastern Siberia (cf. Fig.2), the taiga-tundra key transition zone was confined to the north-central Siberian

region. For comparison, representative pollen records covering the last 6000 years have been chosen..." In the west-central Asian region, the biome model results also deviate strongly from the reference.

R3: "L 254. If possible I would like to see the Figure 4 information using the 0 ka simulated biomes and reference data. Which is a larger differences of biomes between 6 ka/ vs. 0 ka and 0 ka vs. reference data? "

A: Comparing 6ka vs 0ka with 0ka vs reference data is possible, but one can not draw any conclusion from this exercise. The Asian orography is very complex and is only poorly resolved in Earth System models (3.75° on a Gaussian grid). For instance, the Tibetan Plateau is a small 'hill' reaching 4000m in a few grid-boxes, only. The approach of using climate anomalies has the advantage of preserving regional climate pattern that are an imprint of the orography. With this approach, we can use a much finer resolution (0.5°), the resolution of the reference dataset. The difference between 0ka and the reference data might be of similar magnitude as the 6ka-0ka difference, but this is to a large part related to the differences in resolution.

R3: "L 271-272. The results from the sensitivity experiments show the real vegetation response or just BIOME4 response? If we use different vegetation models, do we get different results, for example cloud cover is a key factor of vegetation changes (it is opposite to L 275)? Nemani et al. (2003, Science) also shows the similar results about geographic distribution of potential climatic constraints to plant growth."

A: This is an interesting question. Our results show the response of the BIOME4 model, but the temperature limitation of vegetation in the high northern latitudes and moisture limitation in steppe-desert transition zones are in line with observations and other studies, so, we think that the main results will not change when using other vegetation models.

R3: "L 357. About Figure 11, less/more what? What does the x-axis shows, fractional changes in biome or ...? "

A: As mentioned in the Figure caption, the reconstructions are given in arbitrary units. We now write: "The reconstructions are given in arbitrary units and show the dominant biome at a certain time and the trend qualitatively (i.e. less or more dominant than other biomes)"

R3: "L 357. Why do you use only two reconstructions? Can you use more pollen data from Neotoma database (http://www.neotomadb.org)? "

A: Comparing the simulated biome distributions with (spatial) reconstructions is not the main aim of this study. We are currently preparing a new synthesis of Palaeovegetation change in Asia, which will also be biomised (for China cf. Ni et al., 2014: Biome distribution over the last 22,000 yr in China, doi:10.1016/j.palaeo.2014.04.023). We will compare these datasets with vegetation model results in a follow-up manuscript. We further specify the main aims of this study in the introduction: "The main aims of this study are: a) to get a consistent ensemble of possible changes in biome distribution since the mid-Holocene, b) to test the robustness of the simulated vegetation changes and quantify the differences among the models, i.e. to assess how large the vegetation variability is that results from different climate forcings, and c) to compare simulated vegetation changes in selected key regions with pollen-based reconstructions."

R3: "L371. "The overall change in biome composition since the mid-Holocene"? for "The overall change in the Holocene biome composition" "

A: done.

R3: "L 380-383. You should consider the vegetation model deficits too. Your results really depend on BIOME4 vegetation model."

A: We fully agree, that the results depend on the BIOME4 model. The discussion of the Biome4 model deficits are included in the discussion of the reference simulation (Ch. 2.2).

R3: "L 398. "Local"? for "regional" "

A: done.

R3: "L 405-412. Even if vegetation reconstructions for Asia are sparse, you should use available data (i.e. BIOME 6000 data) for your simulated biomes at the pre-industrial and mid-Holocene first."

A: See comments above, done for mid-Holocene in the Appendix.

R3: "L 410-412. Please put any references."

A: done.

R3: "L 431. Your simulations focus on the last 6000 years, not "during the Holocene" "

A: done.

R3: "L 439. "500 years" for "500years" "

A: done.

R3: "L 496. "More pollen records are needed to evaluate the simulated results." I understand your argument, but please use available dataset and show us the results first."

A: We see the point. We further specify this statement to: "More pollen records are needed to evaluate the simulated results, particularly in the desert-steppe transition zone."

---

## Author Response (AR2)

Referee #2:
The authors have addressed most of the comments raised in previous round in the revised version of the manuscript. There are, however, a few concerns that require the authors to clarify before a final publication:

(1) For all the climate variables used by BIOME4, the authors use "absolute difference" between the climate of each time-slice and the pre-industrial climate, and add the "absolute difference" to the reference present-day climate to generate climate data for each time-slice. This method works well for some variables (e.g., temperature), but can be problematic for other variables (e.g., precipitation). Precipitation can be minus values, if "absolute difference" is too negative. Have the authors checked if there are any problems with adding "absolute difference" to different climate variables? How sensitive are the BIOME4 model results to the ways of adding "anomalies"? More explanations is needed to confirm if the "absolute difference" approach is valid for all variables.
A: This is indeed not specified in the manuscript. Using this anomaly approach can lead to negative values for precipitation and sunshine. In case negative values occurred, we set them to zero.
We added this information to the manuscript and now write: „Negative values in precipitation or sunshine resulting from a too large negative difference of the variables between 6k and 0k, compensating the present-day values, have been set to zero."
This method is a common method used in biome-model studies, see e.g. Wohlfahrt, J., Harrison, S.P., Braconnot, P. et al. Clim Dyn (2008) 31: 871. doi:10.1007/s00382-008-0415-5. or Wohlfahrt, J., Harrison, S.P. & Braconnot, P. Climate Dynamics (2004) 22: 223. doi:10.1007/s00382-003-0379-4.

(2) Fig. 1: Please add "(GPCP dataset for precipitation, ERA40 for wind vectors)" instead of "(based on reanalysis data)".
A: done.

(3) Fig. 11: Time axis of the figure is still not changed. "surrounding of both sides" should be "surrounding of both sites"? The region (106°E-110°E, 69-73°N) selected for the southern Taymyr record does not surround the locality of the record (72.4°N, 102.29°E). The reason for this is not clear to me. It is said that "the modern taiga-tundra margin is shifted westward in the modern biome simulation". According to this, the region selected should also be shifted westward. But the authors selected a region (106°E-110°E, 69-73°N) , which in fact, is to the east of the proxy locality. Why?
A: Thank you, this is indeed a mistake, the Taiga-tundra margin is shifted northward. We simply take the nearest grid-box around the lake showing tundra vegetation in the modern climate. We change this information to: „The latter region had to be adjusted to the modern taiga-tundra margin that is shifted northward in the modern biome simulation due to an overestimation of forest cover in the boreal region. For comparison, the nearest grid-box around the lake showing  tundra vegetation in modern climate has been taken."
Regarding the time axis, we have changed the labeling, but unfortunately put an old version of the figure into the revised manuscript. We now changed it.

Referee #3
Review for "Biome changes in Asia since the mid-Holocene – an analysis of different transient Earth system model simulations"

This manuscript describes the vegetation changes in the Monsoon-Westerly (forest-steppe-desert) transition zone and high northern latitude (taiga-tundra) transition zone in Asia since the mid-Holocene using several different transient simulations and a slightly modified BIOME4 vegetation model. To validate the simulated vegetation changes in the two areas, the authors also try to implement model-data comparison.

The analyses applied to this study are not new (in particular vegetation changes between 0k and 6ka), and thus there are still some problems that should be resolved for data-model comparison with regard to paleo-vegetation changes. Because the paleoclimate and paleovegetation community already know the arguments about data-model comparison in this manuscript, the authors might have to propose a new approach. Anyway, the authors did a great job in analyzing so much data and in producing very nice figures.

The topic is really interesting, but I had several problems here.

Specific comments

L95-98. Although I understand the three research aims in this study, I am not sure that the authors use reasonable approaches for achieving the goals. Data-model comparison approach in this study is not new, and thus we might not improve our knowledge about uncertainties of simulated vegetation changes?
A: The major aim of this study is to provide a range of possible biome changes in East Asia during the last 6000 years. We agree with the Referee that the general methods used in this study are not new, but – at least to our knowledge - a multi-model analysis has not been undertaken for Asia so far, nor transient simulations have been used before. Using transient simulations enables us to analyze trends in biome change. Furthermore, we improved the BIOME4 model for the Asian region.

It seems that the authors mainly use multi-model ensemble approach for the transient vegetation change analyses (Figure 6 - 11), but the authors show the results from the several simulations including the ensemble mean for mid-Holocene vegetation changes (Figure 3 and 5). If possible, you show the results from the ensemble model mean for general trends in main text and each model result in the Appendix because you do not discuss the climate and vegetation changes and their causes among the models thoroughly.
A: We keep the figures 3 and 5 as they are, because we want to provide a range of possible biome distributions for the mid-Holocene time-slice and want to stress that the major vegetation changes and the climate factors yielding the vegetation change are robust, i.e. similar in all models. Mid-Holocene is a key time-slice in the PMIP analysis and, therefore, of special interest for model evaluation and the reconstruction community as synthesis of reconstruction and many climate simulations exist for comparison.
The causes of the differences between the models can not be answered thoroughly without further model experiments.

L105. "absolute minimum temperature" for "annual minimum temperature"?
A: done.

L112-113. The authors need many PFTs and thus biomes for describing the diverse taxa found in Asia. On the other hand, the authors grouped the BIOME4 28 biomes into 12 mega-biomes for the

analysis. It sounds that your argument and your approach is inconsistent here. Moreover, the authors discuss the vegetation margin analysis using only "desert", "steppe", "forest" and "tundra" later. If you use these limited biomes, you can use other process-based vegetation models such as LPJ-GUESS. Your target areas for the vegetation changes in Asia are limited (only transition zones), and the target biomes are also limited.

A: Showing and discussing the change in all 28 biomes used in BIOME4 would result in a very long manuscript. We therefore decided to follow the approach of Harrison and Prentice (2003) to group the biomes in mega-biomes. This has also the advantage that one eliminates indication of small model biases, for instance, when the model simulates biome types which are not the same, but only similar to the observed ones (e.g.cold instead of cool mixed forest). Using only macro-biomes in the 'transient' biome change is also a matter of representation and reducing model biases. But nevertheless, the advantage of using BIOME4 is that all individual biomes are calculated by the model and their climatic tolerance and other requirements are included in the calculation. They are grouped afterwards. Therefore, BIOME4 can respond more specific to the climate forcing than other models, that only use desert, forest and shrubs as vegetation types, and we wanted to calculate biome distributions as pollen-based reconstructions are often assigned to biomes.

(Harrison, S. P., and Prentice, I. C. (2003). Climate and CO2 controls on global vegetation distribution at the last glacial maximum: analysis based on palaeovegetation data, biome modelling and palaeoclimate simulations. Global Change Biology 9, 983-1004.)

L120-123. Although I checked the Table 1 for bioclimatic limits for the model, I do not understand how the authors used the altitude information from ETOPO5. If possible, you should describe how to use the information? Moreover, do we need to change the surface air pressure for running BIOME4 at high altitude grid points?

A: The bioclimatic limits are switched to the „altitudinal vegetation constraints" in grid-boxes with orography higher than 2000m. We added this information in the caption of Table 1: „The limits in the brackets represent the climatic tolerance on mountains (altitudinal vegetation constraints) to which the model switches in grid-boxes with orography exceeding 2000 m."

Regarding the surface air pressure: we looked into the model code and could not find any equation in which the surface air pressure is used.

L134-135. What is 0k and 6k here? You need to define these words before this sentence or here. Otherwise, you can delete the texts after "(Fig. A2)".

A: We changed 0k to pre-industrial and 6k to mid-Holocene.

L138-141. In the previous studies with BIOME4 for simulating paleovegetation, they used CLIMATE2.2 climate data. Why do use CRU TS3.10, not CLIMATE2.2 in this study? If you need higher-resolution of modern climatology as reference data, why do not you use CRU CL2.0 (New et al., 2002)? The data includes the necessary climate variables for BIOME4 and elevation information.

A: We took the newest climate dataset available for us.

L 144-145. To simplify/ignore the impact atmospheric CO2 concentration on vegetation, the authors might use 280 ppm for 0k simulation. If so, you can delete this sentence and the related figure because you do not need to discuss the atmospheric CO2 effect to vegetation in this study.

A: We included this information and the figure at the suggestion of Referee #1 in the first revision. We keep this in the manuscript.

L 151-152. I do not understand what "the main biome distribution" is. Do you mean that the model sufficiently simulates the large-scale (or continental scale) biome distribution in Asia.

A: This is indeed imprecise. We change it to 'large-scale biome distribution'.

L156-168. These sentences describe results and your arguments. Thus you should not put these sentences here.
A: The entire chapter 2.2 deals with model evaluation for pre-industrial climate, it contains methods as well as the description and discussion of the reference simulation. We do not interpret this as results of our study, but rather as detailed description of the model. So we keep these sentences in this chapter.

L179. Remove "of the year"? This absolute minimum temperature is based on an annual data or climatology (e.g. 1960-2000)?
A: done.

L225. "CRU TS3.10" for "CRUTS3.10"
A: Thank you, we changed it.

L229. "(Harrison et al., 1998)" for "(Harrison, 1998)"
A: Thank you. Done.

L230-231.According to Harrison et al. (1998), the anomaly approach has two advantages; 1) reducing the effects of systematic model biases and 2) capturing some of the locale-scale spatial pattern because of terrestrial geography. The authors already describe these advantages in this part, and I understand them. But I do not understand the sentence "We are however aware of… meso- and large scale" Do you need this sentence and the next sentence in this section? I mean, you can describe the methodology here.
A: We rephrase these sentences more precisely and now write: „We are however aware of the simplifications inherent to this approach in interpolating coarsely resolved GCM output (resolution approx. 3.75° and coarser) onto higher target grids (here 0.5°) without taking into account potentially important factors that lead to local variations in climate, such as changes in variability and feedbacks from the local to the meso- and large scale."

L243-244. For comparison, the authors choose one high quality pollen record at each key transition area. But the model output is based on 500-year interval data. Thus, if possible, the authors should use relatively low temporal resolution pollen data. I think one data at each zone is very small number, they might not validate your results well.
A: We agree that higher data coverage would support our results better than using only one record. We have discussed this comment already in the first revision (c.f. author response to Referee #3). High quality data is quite few.  Therefore, we decided to select from all (for us) available records in the target-regions the record, that shows the best quality with respect to dating and data and that is additionally representative for the regional vegetation changes, as inferred from comparison to literature results. The general vegetation trend indicated by the Dahai record is in line with other records in North central China (cf. Zhao et al, 2009) and the 13-CH-12 record agrees with the other records from the northern Siberian tree line area.

L260-272. As mentioned earlier, the authors can describe results from only ensemble model mean because "the vegetation change is small and similar for all models". You can move the other models output into the Appendix. Moreover, You can combine two figures (Figure 2 and Figure 3) into one because you do not need to show "CRU TS3.1 0k" twice.
A: We decided to keep the presentation and discussion of the mid-Holocene (6k) biome distributions simulated by the individual models in the main text, because one aim of the study is to show a range of possible mid-Holocene biome distributions. This is usually the time-slice compared with reconstructions as high-quality synthesis of reconstructions exist for 6k. We also keep the CRU

TS3.1 0k map in both plots, to facilitate the comparison with the reference map and the 6k biome changes.

L282. Delete "East"? Your target area is the entire Asia, not East Asia right?
A: We changed this to Eastern Asia, to be consistent with the other sections.

L283-289. This paragraph (except the first sentence) describes a method for a sensitivity study, not results. Thus, you should move this content to the "Methods" section.
A: We agree, this paragraph rather describes the method of the sensitivity experiment and not results, we move this paragraph to the method section (L238-243) and shorten the information in this paragraph to: „The Holocene changes in bioclimate are discussed in the Appendix (Fig. B1). To assess the climate variables being responsible for the biome shifts in the model since the mid-Holocene, climate variables of the pre-industrial input dataset are gradually replaced by the respective variable of the simulated mid-Holocene climate."

L288. "CRU TS3.10" for "CRUTS3.10"
A: done.

L288. "absolute minimum temperature" for "absolute annual minimum temperature"
A: done.

L290-298. This result is really interesting for me, and it is easy to know the causes of vegetation changes over each transition during the mid-Holocene. Do you make the map from ENSMEAN for Fig.5? Moreover, this information is directly concerned with vegetation changes between 0k and 6k, and thus you can put this information into 3.1.
A: We have not prepared a map based on the ensemble mean climate since the results of the individual simulations are very similar.
We decided to discuss the climate factors in a separate section (3.2).

L291. Your sensitivity experiment cannot deduce "the prolongation of the warmer season" because these climate variables does not have temporal concept probably. We might deduce warmer during the growing season in the northern latitudes (from the table 1: tundra vs. boreal forest and Fig. 5).
A: We are not sure if we understand this comment correctly, we infer the prolongation of the warm-season from the increase in growing degree days.

L301. "46%" in the text, but "45%" in the table.
A:Thank you very much, it is 45%.

L313-. Although "geographical distances are given in degrees of latitude or longitude in the following, for simplification, only ° is used", the authors use "100 km" here. Should you use the unit ° here? The same issues are on L507, L520, and L521.
A: We agree, this is misleading. We did not want to exclude other units, so we change this sentence to: „(for simplification, only ° is used for geographical distances given in degrees of latitude or longitude in the following)."

L375. Is it possible to compare between affinity scores (reconstruction) and fractional area cover (%) of biomes. I hope you understand the concept of biomization.
A: We compare the reconstructions and the simulated biome fractions only semi-quantitatively, i.e. both methods indicate the dominant biome type but tentatively also changes in the relative importance of sub-dominant biome types. But of course affinity scores provide no information about past coverage changes.

Discussin. The authors focus on the two point of discussion. I think you have several arguments in the methods and results sections. Moreover, to support your argument/discussion, you should use other references effectively.

A: We agree, that we discussed the model and the reference simulation in the method section, but we interpret this as evaluation of the model and not as discussion of our results. Therefore, we keep the structure.

We also agree, that there is an imbalance between citations of own studies and other references in chapter 4.1. We add references in the discussion about the impact of differences in interactive model components to the Holocene climate change:

L433-435: „Previous modeling studies indicate that land-surface feedbacks with the atmosphere could have enhanced the Holocene precipitation change induced by orbital forcing (e.g. Wang, 1999; Texier et al., 2000, Diffenbaugh and Sloan, 2002; Li et al., 2009)."

L398. I do not know whether your approach is correct or not. At least, using several transient climate model output, you run the BIOME4, and then you get the simulated biome over the last 6000 years. But, I do not think this small number of output does not show a range of possible biome changes in Asia.

A: We used all (for us) available transient climate simulations, so far, this analysis of transient biome change is all that can be provided by the climate modeling community.

L398-435. I do not understand your discussion (4.1) well. Does your discussion directly have a concoction with your result (which ones)?

A: The simulated biome distributions depend on the climate input. Differences in the simulated climate between the individual simulations lead to differences in the biome distributions. Therefore, we assess the performance of the models in simulating the Asian climate and discuss the factors (differences in model setup) that may lead to differences in the climate. We further explain this in the revised manuscript:

L403-406: „The outcome of this analysis and its reliability depends directly on the capability of the models to simulate the mid-Holocene to pre-industrial climate change. Biases in the models and differences in the simulated climate between the individual models lead to errors and differences in the simulated biome distributions. "

L442-445. I understand your argument here, but you need a limited number of biomes in this study. Moreover, Basil Davis (Université de Lausanne) has a funded project, HORNET (Holocene pollen based climate reconstruction for the Northern Hemisphere extra-tropics) and he makes biomised biome data over the Holocene, so far as I know. Thus, your argument will be solved soon, and you might use his database probably.

A: Thank you very much for the information, we are also in contact with the PAGES LandCover6k initiative and work on the biomisation of pollen-based vegetation reconstructions in several projects, but this is ongoing work.

Table 4, 5, and 6. Do you need to show the all model results here? If you describe the general/common/robust pattern, you can show only 0k and 6k (minimum, ENMEAN, maximum) values without model names.

A: We prefer to show all model results in these tables, as we discuss the positions of the desert-steppe, steppe-forest and taiga-tundra border simulated by the individual models in the Appendix and do not want to show the table twice.

Figure captions. "CRU TS3.10" for "CRUTS3.10" and "BIOME4" for "Biome4"

A: Thank you! We changed it accordingly.

Figure 11. Please put (a), (b), and unit of model output on the figure.
A: We put a) and b) on the figure and added to the caption: „The reconstructions are given in fractional coverage per area [%]."